# AdaDim: Dimensionality Adapation for SSL Representational Dynamics

## Abstract

A key factor in effective Self-Supervised Learning (SSL) is preventing dimensional collapse, where higher-dimensional representation spaces ($R$) span a lower-dimensional subspace. Therefore, SSL optimization strategies involve guiding a model to produce $R$ with a higher dimensionality ($H(R)$) through objectives that encourage decorrelation of features or sample uniformity in $R$. A higher $H(R)$ indicates that $R$ has greater feature diversity which is useful for generalization to downstream tasks. Alongside dimensionality optimization, SSL algorithms also utilize a projection head that maps $R$ into an embedding space $Z$. Recent work has characterized the projection head as a filter of noisy or irrelevant features from the SSL objective by reducing the mutual information $I(R; Z)$. Therefore, the current literature's view is that a good SSL representation space should have a high $H(R)$ and a low $I(R; Z)$. However, this view of SSL is lacking in terms of an understanding of the underlying training dynamics that influences the relationship between both terms. For this reason, we directly oppose the current literature's view of SSL representation spaces and instead assert that the best performing $R$ is one that arrives at an ideal balance between both $H(R)$ and $I(R; Z)$. Our findings reveal that increases in $H(R)$ due to feature decorrelation at the start of training lead to a correspondingly higher $I(R; Z)$, while increases in $H(R)$ due to samples distributing uniformly in a high-dimensional space at the end of training cause $I(R; Z)$ to plateau or decrease. Furthermore, our analysis shows that the best performing SSL models do not have the highest $H(R)$ nor the lowest $I(R; Z)$, but effectively arrive at a balance between both. To take advantage of this analysis, we introduce AdaDim, a method that leverages SSL training dynamics by adaptively balancing between increasing $H(R)$ through feature decorrelation and sample uniformity as well as gradual regularization of $I(R; Z)$ as training progresses. We show that AdaDim results in performance exceeding common SSL baselines without necessitating expensive architectural strategies. However, in settings where we integrate these techniques, we demonstrate even further performance gains exceeding the state of the art in standard benchmark tasks.

## 1 Introduction

Self-supervised Learning (SSL) (48) algorithms approach or surpass fully supervised strategies on a wide variety of benchmark tasks (8; 7; 15; 56; 3; 9). SSL optimization generally involves an invariance loss that ensures representations of similar samples align with each other and a mechanism to prevent dimensional collapse (22). Dimensional collapse refers to the phenomena where high dimensional representations span a lower-dimensional subspace. Therefore, to prevent dimensional collapse, a wide variety of works (18; 2; 45) suggest that good SSL representations ($R$) should have a higher overall dimensionality. These works arrive at this conclusion through some measurement of the uniformity of the eigenvalue spectrum of a matrix derived from its representation space. In this work, we analytically measure the dimensionality of the representation space $H(R)$ through the effective rank metric (38). Effective rank quantifies the entropy of singular values of $R$ and provides a matrix approximation of entropy (50; 37). It is for this reason that we refer to dimensionality as $H(R)$ since eigenvalues approaching a uniform distribution reflect the spread of samples along higher dimensional feature directions.(Additional discussion and details of all metrics can be found in Section A.6.) In practice, optimizing for higher dimensionality is either done through a dimension contrastive approach

(19) that encourages feature decorrelation or through a sample-contrastive method that promotes a uniform spread of sample representations (51). Alongside a term to promote dimensionality, all SSL methods utilize a projection head that maps $R$ into a lower dimensional embedding space $Z$ where the SSL optimization objective is applied. Recent work (35) has characterized the purpose of the projection head as a filter that removes spurious features thus lowering the mutual information $I(R; Z)$ that we measure through a matrix entropy approximator (37; 58). In general, lower $I(R; Z)$ reflects representations varying only in feature directions that correspond well with task-relevant semantic concepts, while higher $H(R)$ corresponds to a greater degree of feature diversity. Together, these works imply that a good SSL representation space should have a high $H(R)$ and low $I(R; Z)$.

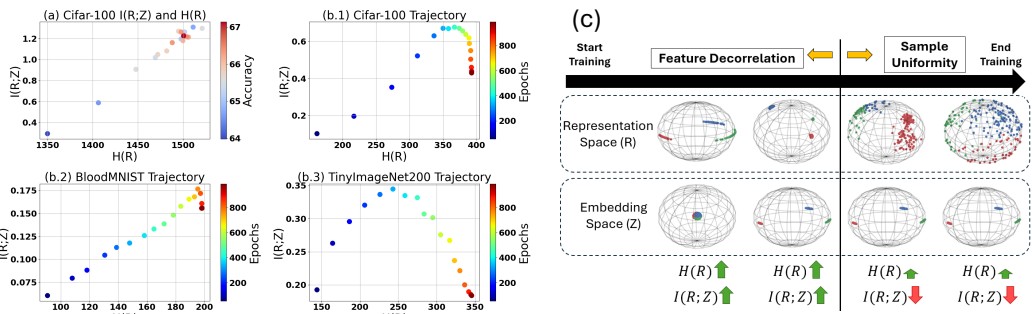

Figure 1: a) This figure shows how performance varies for 20 different pre-trained ResNet-50 models as a function of $H(R)$ and $I(R; Z)$. b.1) - b.3) shows how $H(R)$ and $I(R; Z)$ vary across training of a ResNet-18 encoder with SimCLR (7) for 1000 epochs on three different datasets. c) This toy graphic shows how the representation space ($R$) and embedding space ($Z$) of a toy 3D dataset changes when following SSL training dynamics. We also visualize the impact on $H(R)$ and $I(R; Z)$.

However, this view of SSL is lacking in terms of an understanding of the underlying training dynamics that influences the relationship between both terms. For example, in part a) of Figure 1, we show how the final $H(R)$ and $I(R; Z)$ arrived at the end of training influences downstream performance. In this Figure, we train 20 different models with slightly different hyperparameters with a ResNet-50 (21) model for 400 epochs on Cifar-100. We find that the best performing model is not the one with the highest $H(R)$ or lowest $I(R; Z)$, but instead approaches a specific $H(R)$ and $I(R; Z)$ value where downstream performance is maximized. **Thus, our first claim is that the best performing SSL representations arrive at a balance between both $H(R)$ and $I(R; Z)$ such that there is enough feature diversity for the task of interest, but not so much that $R$ contains irrelevant noise.** This claim directly opposes existing literature (18; 2; 45) that only considers some associated measurement of the $H(R)$ reached at the end of training as an indicator of downstream model performance.

In this work, we also analyze the representational dynamics that cause this behavior. In parts b.1) - b.3) of Figure 1 we show how $H(R)$ and $I(R; Z)$ evolve over the course of SimCLR (7) training on a ResNet-18 model for 1000 epochs across 3 distinct datasets. While $H(R)$ generally increases throughout training, as expected by the current literature, $I(R; Z)$ does not directly decrease and instead goes through distinct phases of increasing, plateauing, and decreasing. In part c), we show a toy example to visualize the dynamics causing this behavior. In this Figure, we have 200 samples distributed within a fictitious 3D spherical representation space. At the start of training, $H(R)$ increases by projecting $R$ onto a higher dimensional space by mapping from a 2D plane to the surface of the sphere. $Z$ correspondingly projects from a 1D to 2D space. This phase corresponds to feature decorrelation where both $R$ and $Z$ increase the number of dimensions in which they vary which causes $I(R; Z)$ to increase as both spaces are projecting to a higher dimension. However, later in training, $H(R)$ has fewer dimensions in which to project into and further increases in $H(R)$ are caused by samples distributing uniformly within the space. This change in sample spread is not reflected to the same degree in $Z$ which causes $I(R; Z)$ to decrease. Thus, our **second claim is that feature decorrelation at the start of training leads to higher $I(R; Z)$, while samples uniformly spreading across higher dimensions at the end of training causes $I(R; Z)$ to plateau or decrease.**

Based on our first two claims, we propose an SSL method called AdaDim. AdaDim takes advantage of the discussed training dynamics to adaptively balance increasing $H(R)$ through feature decorrelation and sample uniformity as well as gradual regularization of $I(R; Z)$ as training progresses. This

adaptation is done in a manner that is specific to the dimensionality characteristics of the dataset of interest. This method implies our **third claim which is SSL optimization objectives should be constructed to allow adaptation to evolving $H(R)$ and $I(R; Z)$ dynamics.**

1. We theoretically and empirically demonstrate that the relationship between $H(R)$ and $I(R; Z)$ can characterize SSL training dynamics through both a gaussian and information theoretic analysis.

2. We empirically validate that the best performing SSL models use the discussed dynamics to arrive at an ideal balance for both $H(R)$ and $I(R; Z)$ by the end of training.

3. We develop a dimension adaptive (`AdaDim`) method that exploits our discovered training dynamics to regularize the training process towards balancing both $H(R)$ and $I(R; Z)$. We demonstrate performance improvements in comparison with state of the art methods without needing expensive architectural strategies. However, we also show that our method is also able to leverage these techniques for further performance gains.

## 2 RELATED WORKS

**SSL Methods**   (19) categorizes SSL methods as dimension-contrastive or sample-contrastive. Sample contrastive methods work by projecting sample augmentations (positives) closer to each other than that of other samples in a batch (negatives) (7). Other methods are derived from simple alterations to the definition of positive and negative sets. Research directions include using a momentum queue (8), using nearest neighbors as positives (15), enforcing cluster assignments (5), enforcing hierarchical structures (30; 27), and using label information (24). Dimension contrastive approaches enforce feature decorrelation through various methods. Examples include regularizing the embedding covariance matrix (3; 56; 16) or introducing architectural constraints (9; 20; 6) that implicitly regularize dimensions. Our method differs due to the introduction of an adaptive mechanism to take advantage of the underlying training dynamics of both sample and dimension contrastive approaches at different points of training. We also note that (19) discussed conditions under which both sample and dimension contrastive approaches are equivalent from an optimization perspective. However, the authors of this work also acknowledge that neither approach can be used interchangeably. More recent work (41) identified differences in the entropy characteristics of both approaches and suggested that it may be possible to devise methods that are able to take advantage of the characteristics of both. Therefore, our work can also be understood from the perspective of identifying the underlying dynamics where both approaches contribute meaningfully to SSL optimization.

**Understanding SSL Training Dynamics**   A subset of works has also attempted to understand the training dynamics of SSL methods. (22) analyzed the dimensional collapse phenomenon within contrastive learning settings. (42) explored the idea that SSL training dynamics involves learning one eigenvalue at a time. (46; 43) analyzed the learning dynamics of dimension contrastive methods in the context of simple linear networks. In general, there is much more depth of literature for understanding training dynamics within supervised settings (1; 17; 40), while work into understanding the underlying dynamics of SSL methods is limited. Our work understands SSL through the lens of characterizing training dynamics by the relationship between $I(R; Z)$ and $H(R)$.

## 3 ANALYSIS OF TRAINING DYNAMICS

### 3.1 SIMULATED TRAINING DYNAMICS

Through the analyses of this section, we find that increases in $H(R)$ due to feature decorrelation causes a corresponding increase in $I(R; Z)$ while increases in $H(R)$ due to sample uniformity causes $I(R; Z)$ to plateau or decrease. To investigate these dynamics, we perform a simulation within a Gaussian setting. Assume that Gaussian distributed data is represented by $R \sim \mathcal{N}(\mu_R, \Sigma_R)$ where $R \in \mathcal{R}^m$. Additionally, assume that there is some projection of $R$ represented by $Z \sim \mathcal{N}(\mu_Z, \Sigma_Z)$ where $Z \in \mathcal{R}^n$ such that $n < m$. $R$ and $Z$ form a jointly multivariate normal distribution. Together, this distribution is defined by a block covariance matrix of the form $\Sigma = \begin{bmatrix} \Sigma_Z & \Sigma_{ZR} \\ \Sigma_{RZ} & \Sigma_R \end{bmatrix}$. In this

setting, the closed form solution for $I(R; Z) = \frac{1}{2}(ln(|\Sigma_R|) + ln(|\Sigma_Z|) - ln(|\Sigma|))$. Applying Shur's complement to the block covariance matrix results in the following equation when all covariance matrices are invertible:

$$I(R; Z) = \frac{1}{2}(ln(|\Sigma_Z|) - ln(|Var(Z|R)|)) = \frac{1}{2}(ln(|\Sigma_R|) - ln(|Var(R|Z)|)) \qquad (1)$$

In equation 1, $Var(Z|R) = \Sigma_Z - \Sigma_{RZ}\Sigma_R^{-1}\Sigma_{ZR}$ and $Var(R|Z) = \Sigma_R - \Sigma_{ZR}\Sigma_Z^{-1}\Sigma_{RZ}$. The details of this derivation can be found in Section B.2. From this construction of the problem, several trends emerge. $I(R; Z)$ will increase or decrease depending on the relationship that the projection produces between $R$ and $Z$. Specifically, $I(R; Z)$ will increase when the variance of the space of interest increases while its corresponding conditional variance remains relatively lower. These variance changes can occur through a larger number of features or through a more uniform spread of data samples. Figure 2 demonstrates a simulation of the effect of each by generating a synthetic gaussian dataset with 1000 samples, a defined variance for each of 5 generated clusters, and a defined number of features $m > 10$ to simulate $R$. $R$ is then projected with PCA to generate $Z$ with either 2 components or 10 components denoted by $n$. This design choice is to simulate the difference between early and late stage SSL training. Early in training, $R$ and $Z$ project closer to each other which is represented by the 10 component $Z$ space while later in training $R$ and $Z$ diverge to a greater degree represented by the 2 component projection. Note that a gaussian assumption on the distribution of the data space is standard practice within the analysis of SSL methods (41). Additionally, PCA serves as a representative projection in this setting, since the information content of $R$ is represented by the variance parameter of an $m$-dimensional Gaussian. However, the same simulation is repeated with similar conclusions in Section B.4 where the projector is replaced with a small neural network. Further details of these experiments can be found in Section B.3.

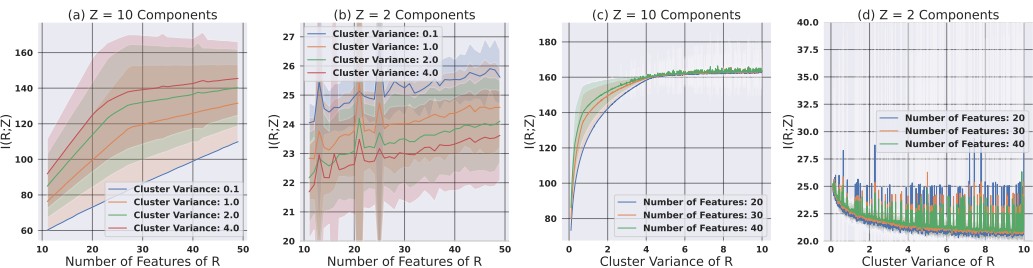

Figure 2: a) and b) show how the $I(R; Z)$ changes as the number of features of $R$ is increased. c) and d) show how $I(R; Z)$ varies as the sample cluster variance increases. Both pairs of Figures show the underlying dynamics under a large projection $Z = 2$ and lesser projection $Z = 10$ PCA components.

In Figures 2 a) and b), $H(R)$ increases by increasing the number of generated features $m$ while the cluster variance is kept constant. This corresponds to the feature decorrelation setting. The second experiment in Figures 2 c) and d) involves changing the sample variance while keeping the number of features fixed which corresponds to the setting where the sample uniformity changes between spaces. In parts a) and b), for different cluster variance values, increasing the number of features in $R$ corresponds to an increase in $I(R; Z)$ regardless of the degree of projection. In parts c) and d), the behavior of $I(R; Z)$ varies significantly based on the degree of the projection. For the 10 component projection case of part c), increasing the sample variance initially increases $I(R; Z)$, but it gradually plateaus as the sample variance increases further. This suggests that the projection cannot capture the variance along certain dimensions after a specific point. In part d), in the 2 component case, increasing the sample variance by any amount reduces $I(R; Z)$. **Overall, this Figure shows that** $I(R; Z)$ **increases with a greater number of decorrelated features in** $R$ **regardless of the degree of the projection. In contrast,** $I(R; Z)$ **increases, plateaus, or decreases based on the degree of sample variance and projection from space** $m$ **to** $n$**.** The exact choice of SSL optimization objective and training procedures will influence the degree to which $H(R)$ and $I(R; Z)$ increases or decreases, but the underlying representational dynamics will reflect our analysis.

Another important consideration is how downstream performance is influenced by the relationship between $H(R)$ and $I(R;Z)$ at the end of training. To model this, the SSL information flow can be described by: $Y \rightarrow X \rightarrow R \rightarrow Z \rightarrow T$. $Y$ represents the semantic concept associated with the data $X$. $T$ represents the associated SSL task. The end goal of the SSL objective is to maximize $I(Y;R)$ which is the mutual information between the semantics of the data and the representation space. Recent work (35) showed that this information flow results in an upper bound on $I(Y;R)$:

$$I(Y;R) \leq I(Y;Z) - I(R;Z) + H(R) \tag{2}$$

Our objective is to show how this bound is effected by the training dynamics discussed in Section 3.1 and to show that simply reducing $I(R;Z)$ and increasing $H(R)$ to maximize this bound is not possible given these dynamics. It is assumed that $R$ and $Z$ are drawn from a joint multivariate Gaussian distribution. Furthermore, $I(Y;Z)$ is assumed to approach some constant $G$ to isolate the analysis with respect to $I(R;Z)$ and $H(R)$. The justification for this term acting as a constant is from previous analyses (39) that assumed the information shared between semantic labels and the target SSL task can be regarded as a constant. Equation 2 can then be rewritten as:

$$I(Y;R) \leq G + \underbrace{\frac{1}{2}(ln(|\Sigma_R|) - ln(|\Sigma_Z|))}_{K(Both)} + \underbrace{\frac{1}{2}ln(|Var(Z|R)|)}_{V(I(R;Z))} + \underbrace{\frac{m}{2}(ln(2\pi) + 1)}_{D(H(R))} \tag{3}$$

Equation 3 suggests that the bound on $I(Y;R)$ can be decomposed into three terms: a variance differential term $K$, a conditional variance term $V$, and a total dimension term $D$. The derivation of this bound is shown in Section B.5. Each term is labeled by its effect on $I(R;Z)$ or $H(R)$. Ideally, increasing each of these terms together would result in a higher overall bound on $I(Y;R)$. However, the SSL training dynamics discussed in Section 3.1 leads to the emergence of a dynamical system where increasing one of these terms can potentially limit the growth of the others. For example, if $H(R)$ increases via feature decorrelation, then $D$ will increase due to a greater number of features $m$. This, in turn, causes feature decorrelation in $Z$, as discussed in our previous dynamics, which causes $I(R;Z)$ to increase and $V$ to decrease which limits the upper bound in equation 3. Additionally, $K$ is limited in this setting due to both of its terms increasing together. In the setting where $H(R)$ increases due to sample uniformity, $D$ is fixed in the number of dimensions which acts as a bound on how large $H(R)$ can grow. In contrast, $K$ and $V$ increase due to an increase in the variance of $R$ without the projection $Z$ having a corresponding increase in variance which lowers $I(R;Z)$. **This oscillatory behavior between each of these terms suggests that the downstream performance represented by $I(Y;R)$ cannot be maximized by optimizing for each term individually and requires a procedure that adaptively finds a balance.**

## 3.2 EMPIRICAL DYNAMICS

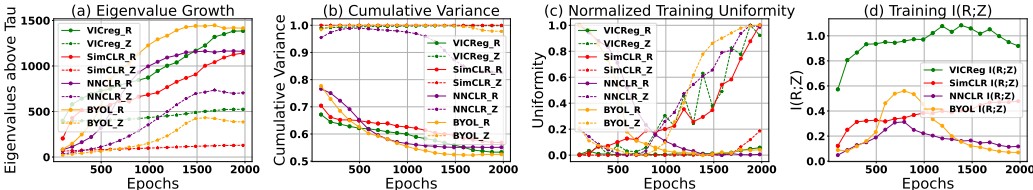

Figure 3: This is an analysis of the $R$ and $Z$ space for 4 different SSL models trained for 2000 epochs on Cifar-100 with ResNet-50. This analysis includes a) the number of eigenvalues above a threshold of $\tau = .01$, b) the cumulative explained variance ratio for the top 30% of eigenvalues, c) the uniformity of each space, and d) $I(R;Z)$.

To verify the theoretical dynamics discussed in Section 3.1, an empirical analysis within a real SSL setting is shown in Figure 3. This experiment involves training a ResNet-50 model (21) with 4 different SSL methods for 2000 epochs on Cifar-100. The projector is designed such that $R$ and

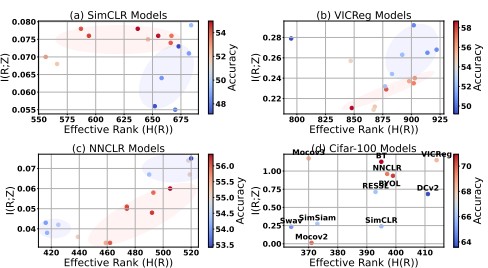

| Magnitude of Correlation with Performance Across Trained Models | | | | | | |
|---|---|---|---|---|---|---|
| Method | Dataset | Epochs | # of Models | H(R) | I(R;Z) | Ratio |
| SimCLR | Cifar100 | 100 | 15 | .082 | .323 | **.462** |
| VICReg | Cifar100 | 100 | 15 | .013 | **.772** | .751 |
| NNCLR | Cifar100 | 100 | 15 | .206 | .229 | **.337** |
| All-ResNet18 | Cifar100 | 1000 | 11 | .029 | .372 | **.375** |
| SimCLR | Cifar100 | 400 | 10 | .557 | .543 | **.625** |
| VICReg | Cifar100 | 400 | 10 | .351 | .875 | **.894** |
| SimCLR | TinyImageNet200 | 400 | 10 | **.534** | .507 | .521 |
| SimCLR | Cinic-10 | 400 | 10 | .029 | .323 | **.421** |
| SimCLR | Cifar-10 | 400 | 10 | **.873** | .841 | .833 |
| SimCLR | OrganSMNIST | 400 | 10 | .0024 | .435 | **.442** |

Figure 4: In Figures a), b), and c), the $H(R)$ and $I(R;Z)$ across 15 ResNet-50 models trained with randomized hyperparameters with 3 different SSL strategies are shown. In Figure d), we show the same plot across 11 different SSL methods trained on ResNet-18 for 1000 epochs each.

Table 1: This table shows the pearson correlation coefficient between the performance of a set of SSL models trained with different hyperparameters on a specific dataset and the effective rank ($H(R)$), $I(R;Z)$, and the ratio between them.

$Z$ both have 2048 features. For all analytical experiments, the matrix being analyzed is the matrix formed by passing each sample from the test set into the encoder network and then concatenating all resulting representations into a single matrix. Details of these experiments can be found in Section A.9. In part a), we analyze the evolution of feature decorrelation for both the $R$ and $Z$ space across training by performing a count of the number of eigenvalues above a threshold $\tau = .01$. It is interesting to note that for the $R$ space the number of eigenvalues consistently increases until late in training while the $Z$ space has a more pronounced plateauing behavior earlier in training. This shows the behavior that the overall dimension of both spaces diverges from each other during training. In part b), we analyze the uniformity of eigenvalues by measuring what percentage of the variance in the space of interest is represented by the top 30% of eigenvalues. This is known as the cumulative explained variance ratio (23). We observe that the cumulative explained variance of $R$ for all methods decreases during training which indicates that $H(R)$ is increasing due to a more uniform spread of eigenvalues and will gradually depend more on sample uniformity as training progresses. However, in $Z$, this metric is near 1.0 for all epochs of training which means that most of the variance of $Z$ is contained within only a small number of top eigenvalues. This suggests that samples in $Z$ distribute uniformly along a restricted subset of dimensions which is in contrast to the behavior of space $R$ that tries to distribute uniformly on as many dimensions as possible. This discrepancy in sample uniformity can also be visualized in part c) with the uniformity metric (51). We observe that for all SSL methods the uniformity between both spaces diverges from each other as training progresses. This divergent behavior is further confirmed in part d), where $I(R;Z)$ increases at the start of training, but gradually decreases for every method later in training.

We also empirically verify how the relationship between $H(R)$ and $I(R;Z)$ impacts the downstream performance in Figure 4. In parts a), b), and c) we train 15 different models with randomized hyperparameters specific to 3 different SSL methods with a ResNet-50 model on Cifar-100 for 100 epochs each. We observe that for each method, the best performing models cluster around specific $H(R)$ and $I(R;Z)$ values. This trend also holds in part d), where all 11 models are trained with entirely different SSL approaches. In Table 1, we compute the magnitude of the Pearson correlation coefficient between the performance of each of the generated models across different datasets and $H(R)$, $I(R;Z)$, and the ratio between both of them. We observe that generally the performance correlates more with the ratio, rather than either of the terms individually. Again, this result empirically shows the existence of an ideal balance between $H(R)$ and $I(R;Z)$ that corresponds to the best performing SSL model.

## 4 METHODOLOGY

Based on the analysis of the previous section, we introduce a method to balance the training trajectory of both $H(R)$ and $I(R;Z)$. Consider an image $i$ drawn from a training pool $i \in I$. $i$ is passed into two random transformations $a(i) = x_i$ and $a'(i) = x'_i$ where $a$ and $a'$ are drawn from the set of all random augmentations $A$. Both $x_i$ and $x'_i$ are passed into an encoder network $e(\cdot)$. This

results in the representations $e(x) = r_i$ and $e(x^{'}) = r_i^{'}$. These representations are then passed into a projection head $g(\cdot)$ that produces the embeddings $g(x_i) = z_i$ and $g(x_i^{'}) = z_i^{'}$. The collection of all representations and embeddings within a batch of $b$ samples can be represented by the $R$, $R^{'}$, $Z$, and $Z^{'}$ matrices. In this case, all matrices are composed of $b$ vectors with $F$ features. From this setup, we can compute $L_{NCE}$ used in SimCLR (7) and the $L_{VICReg}$ (3) loss. Note for embeddings passed into $L_{NCE}$ further normalization is applied on $z_i$ to produce $\hat{z}_i$. The main details of each loss are provided in Section A.5. For the purposes of discussing AdaDim, we highlight the sample uniformity term in $L_{NCE}$ and the feature decorrelation term in $L_{VICReg}$:

$$L_{NCE} = \sum_{i \in I} (-\hat{z}_i \cdot \hat{z}_i^{'})/\tau + \underbrace{log(\sum_{k \in K(i)} exp(\hat{z}_i \cdot \hat{z}_k / \tau)))}_{uniformity} \quad L_{VICReg} = \lambda s(Z, Z^{'}) + \mu[v(Z) + v(Z^{'})] + \underbrace{\nu[c(Z) + c(Z^{'})]}_{decorrelation}$$

$$(4)$$

The second term in $L_{NCE}$ is a sample uniformity loss as it distances the image of interest $\hat{z}_i$ away from all other samples in the batch of interest $k \in K(i)$. The final term in $L_{VICReg}$ represents a decorrelation loss as it tries to drive the covariance matrix towards an identity matrix. It takes the form $c(Z) = \frac{1}{F} \sum_{i \neq j} [C(Z)]_{i,j}^2$ where $C(Z)$ is the covariance matrix of $Z$. We then compute the dimensionality of the current embedding space $Z$ after every $e_\alpha$ epochs (20 in this paper) of training. This is done by computing the SVD of the representation space of 10 randomly chosen batches from the training set and then calculating the average effective rank across these batches $ER(Z)$ (38). We then scale $ER(Z)$ by the maximum possible dimensionality value which is $D = min(b, F)$ to produce the adaptive parameter $\alpha = \frac{ER(Z)}{D}$. $\alpha$ will gradually transition from 0 to 1 during training as the dimensionality of the space increases. Therefore, we can transition between optimizing between feature decorrelation and sample uniformity with the loss $(1 - \alpha)L_{VICReg} + \alpha L_{NCE}$. However, we also want to gradually increase regularization on $I(R; Z)$ to counter the decrease in $I(R; Z)$ that emerges later in SSL training. To do this, we compute an $I(R; Z)$ loss $L_{mut}(R, Z)$ that encourages higher $I(R; Z)$ with the $\alpha$-Renyi entropy approximation technique (58; 35; 37). This loss first computes the entropy of a matrix with the formula $H(R) = -\frac{1}{2}log[tr(\frac{R}{b})^2]$. The mutual information can then be computed as $I(R; Z) = H(R) + H(Z) - H(R \odot Z)$. For the purpose of numerical stability, the $I(R; Z)$ loss is computed as $L_{mut} = I(\hat{R}\hat{R}^T; \hat{Z}\hat{Z}^T)$ where $\hat{R}$ and $\hat{Z}$ refer to the normalized version of each space. We scale its regularization through the term $\beta = \gamma * \alpha$ with $\gamma$ set as a constant. We provide PyTorch style pseudo-code of our method in Section A.3. We also note that this general loss can be applied regardless of additional architectural overhead used in SSL strategies such as a momentum encoder (8) or a predictor head (52). Our final loss is:

$$L_{AdaDim} = (1 - \beta)[(1 - \alpha)L_{VICReg} + \alpha L_{NCE}] - \beta L_{mut} \quad (5)$$

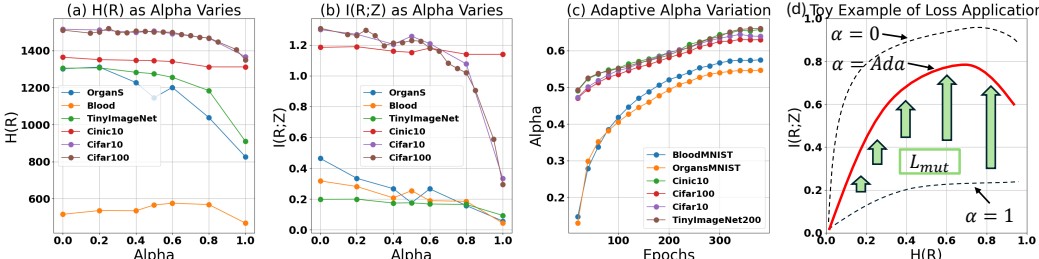

Figure 5: This figure shows the impact of manually varying alpha on a) $H(R)$ and b) $I(R; Z)$ in the setting where $\beta$ is set as a constant 0. c) This figure shows how the adaptive $\alpha$ parameter varies during training of a ResNet50 model for 400 epochs. d) This figure gives a toy example of the intuition behind our loss that includes the adaptive $\alpha$ leading to an intermediate $H(R)$ and $I(R; Z)$ trajectory alongside gradual increases in $L_{mut}$ regularization.

The design of this loss is based on our goal of naturally leading to an ideal balance between $H(R)$ and $I(R; Z)$. To achieve this balance, the loss needs different components that both support and

| AdaDim Parameter Variation Ablation | | | | |
|---|---|---|---|---|
| Method | $\alpha$ | $\gamma$ | $\beta$ | Accuracy |
| AdaDim | 0 | 0 | 0 | 72.14 |
| AdaDim | 1 | 0 | 0 | 69.57 |
| AdaDim | 0.5 | 0 | 0 | 72.23 |
| AdaDim | Ada | 0 | 0 | 72.30 |
| AdaDim | Cosine | 0 | 0 | 71.40 |
| AdaDim | Linear | 0 | 0 | 71.59 |
| AdaDim | Ada | 1e-04 | 1 | 72.00 |
| AdaDim | 1 | 1e-04 | 1 | 68.81 |
| AdaDim | 0 | 1e-04 | 1 | 72.10 |
| AdaDim | Ada | 1e-04 | Ada | **72.73** |
| SimCLR + $\lambda$ (35) | - | - | - | 69.11 |
| VICReg + $\lambda$ (35) | - | - | - | 72.14 |

Table 2: This table studies the effect of variations to the $\alpha$, $\beta$, and $\gamma$ parameters on performance. Experiments use a ResNet-50 model on Cifar-100 for 400 epochs with the baseline hyperparameter setting.

| AdaDim Diverse Dataset Comparison | | | | | | | | |
|---|---|---|---|---|---|---|---|---|
| Method | Cifar100 | TinyImageNet200 | Cinic10 | STL10 | Blood | OrganA | OrganS | OrganC |
| SimCLR (7) | 69.06 | 46.66 | 78.77 | 86.73 | 93.10 | 88.04 | 77.98 | 91.13 |
| VICReg (3) | 72.18 | 48.47 | 82.70 | 87.92 | 93.77 | 92.21 | 80.37 | 91.84 |
| Moco v2 (8) | 71.01 | 46.78 | 81.48 | 92.41 | 93.74 | 90.49 | 75.96 | 90.81 |
| BYOL (20) | 71.72 | 32.96 | 80.00 | 89.96 | 92.45 | 92.26 | 78.53 | 91.45 |
| Barlow Twins (3) | 70.84 | 46.73 | 81.5 | 88.45 | 89.91 | 91.69 | 78.69 | 89.77 |
| NNCLR (15) | 70.72 | 39.66 | 77.28 | 87.16 | 93.15 | 92.93 | 79.92 | 91.71 |
| SimSiam (9) | 65.52 | 31.35 | 79.97 | 89.45 | 91.78 | 91.91 | 78.31 | 90.79 |
| Deepcluster v2 (4) | 65.70 | 41.87 | 74.80 | 82.93 | 93.56 | 92.21 | 77.93 | 74.31 |
| Moco v3 (6) | 63.96 | 37.56 | 74.71 | 85.25 | 93.33 | 92.27 | 78.69 | 91.84 |
| AdaDim ($\gamma = 0$) | 72.23 | 47.87 | 82.38 | 88.11 | 93.74 | 92.90 | 80.19 | 91.95 |
| AdaDim ($\gamma = 1e-4$) | 72.73 | 48.77 | 82.77 | 89.01 | 94.24 | 92.77 | 80.80 | 91.95 |
| AdaDim ($\gamma$=Tuned) | 72.73 | 48.76 | 82.84 | 89.21 | 94.24 | 93.34 | 80.80 | 92.33 |
| Resa (52) | 72.06 | 50.73 | 84.23 | 92.56 | 88.71 | 93.20 | 78.81 | 91.50 |
| AdaDim + momentum | **75.38** | **54.68** | **84.99** | 92.70 | **95.21** | **93.67** | 78.83 | 91.82 |

Table 3: Comparison methods use the given parameters from (11). Experiments involve a ResNet-50 model for 400 epochs with baseline hyperparameters as well as comparisons with an additional momentum encoder and the Resa method (52) under the expanded hyperparameter setting. For the AdaDim baseline setting, we vary $\gamma$ to 0, 1e-4, and a value tuned to specific datasets.

oppose the growth of $H(R)$ and $I(R; Z)$ at different points during the training process by exploiting the observed dynamics that we discuss in Section 3. The first set of components that are balanced with the $\alpha$ term are $L_{NCE}$ and $L_{VICReg}$. In parts a) and b) of Figure 5, we show the impact on $H(R)$ and $I(R; Z)$ when manually varying $\alpha$ from 0 to 1 while fixing $\beta = 0$ across 6 different datasets. As a loss based on sample uniformity, $L_{NCE}$ supports lower $H(R)$ and $I(R; Z)$ while a feature decorrelation based loss like $L_{VICReg}$ supports higher $I(R; Z)$ and $H(R)$. This leads to the behavior of parts a) and b), where gradually varying the loss from 0 ($L_{VICReg}$) to 1 ($L_{NCE}$) consistently leads to both a lower $H(R)$ and $I(R; Z)$. In part c), we show that the adaptive $\alpha$ term grows from 0 to 1 in a manner that is specific to the unique dimensionality characteristics of each dataset. Therefore, the adaptive $\alpha$ term encourages an intermediate $H(R)$ and $I(R; Z)$ training trajectory when compared with $\alpha = 0$ or $\alpha = 1$ as shown in the toy intuition example of part d). However, at the end of training, both $L_{VICReg}$ and $L_{NCE}$ will demonstrate the SSL dynamic of lowering $I(R; Z)$ in late stage training. Therefore, to maintain balance in this dynamic system, we need an additional term that explicitly opposes the decrease in $I(R; Z)$ as shown by the magnitude of $L_{mut}$ increasing as it scales with $\alpha$ in part d). In this way, AdaDim dynamically balances both $H(R)$ and $I(R; Z)$.

# 5 RESULTS

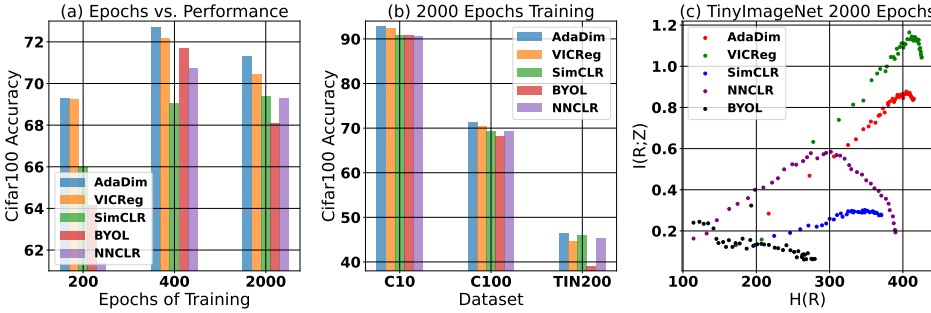

Figure 6: a) This figure demonstrates the performance of AdaDim under different amounts of training epochs on Cifar-100 with a ResNet-50 model for 200 and 400 epochs and a ResNet-18 model for 2000 epochs. b) This figure shows how AdaDim performs under a long training setting of 2000 epochs with a ResNet-18 encoder for Cifar-10 (C10), Cifar-100 (C100), and TinyImageNet200 (TIN200) c) This shows the $H(R)$ and $I(R; Z)$ trajectory over 2000 epochs with a ResNet-18 encoder.

To thoroughly analyze AdaDim, we introduce two different sets of hyperparameters: the **baseline and expanded settings**. In the baseline setting, we use the traditional joint embedding setup with a single ResNet-50 (21) backbone in tandem with a simple 3-layer MLP projector that has an output

| Solo-Learn (11) Benchmark Comparison | | | | | | | |
|---|---|---|---|---|---|---|---|
| Method | M | P | Q | C | Cifar-10 | Cifar-100 | ImageNet-100 |
| Barlow Twins (56) | ✗ | ✗ | ✗ | ✗ | 92.10 | 70.90 | 80.38 |
| BYOL (20) | ✓ | ✓ | ✗ | ✗ | 92.58 | 70.46 | 80.16 |
| Deep Cluster v2 (4) | ✗ | ✗ | ✗ | ✓ | 88.85 | 63.61 | 80.32 |
| DINO (6) | ✗ | ✗ | ✗ | ✗ | 89.52 | 66.76 | 74.84 |
| Moco v2 (8) | ✓ | ✗ | ✓ | ✗ | 92.94 | 69.89 | 78.20 |
| Moco v3 (10) | ✓ | ✓ | ✗ | ✗ | 93.10 | 68.83 | 80.36 |
| NNCLR (15) | ✗ | ✓ | ✓ | ✓ | 91.88 | 69.62 | 79.80 |
| ReSSL (59) | ✓ | ✗ | ✓ | ✗ | 90.63 | 65.92 | 76.92 |
| SimCLR (7) | ✗ | ✗ | ✗ | ✗ | 90.74 | 65.78 | 77.64 |
| SimSiam (9) | ✗ | ✓ | ✗ | ✗ | 90.51 | 66.04 | 74.54 |
| SwAV (5) | ✗ | ✗ | ✗ | ✓ | 89.17 | 64.88 | 74.04 |
| VICReg (3) | ✗ | ✗ | ✗ | ✗ | 92.07 | 68.54 | 79.22 |
| AdaDim (Ours) | ✗ | ✗ | ✗ | ✗ | 92.81 | 71.20 | 80.78 |
| Resa (52) | ✓ | ✗ | ✗ | ✓ | 93.53 | 72.21 | 82.24 |
| AdaDim (Ours) | ✓ | ✗ | ✗ | ✗ | 93.61 | 74.31 | 83.10 |

| ImageNet SOTA Comparison | | | | | | |
|---|---|---|---|---|---|---|
| Method | M | P | Q | C | Batch Size | Accuracy |
| SimCLR (7) | ✗ | ✗ | ✗ | ✗ | 4096 | 66.50 |
| SwAV (57) | ✗ | ✗ | ✗ | ✓ | 4096 | 66.50 |
| Moco v3 (10) | ✓ | ✓ | ✗ | ✗ | 4096 | 68.90 |
| BYOL (20) | ✓ | ✓ | ✗ | ✗ | 4096 | 66.50 |
| Barlow Twins (56) | ✗ | ✗ | ✗ | ✗ | 2048 | 67.70 |
| VICReg (3) | ✗ | ✗ | ✗ | ✗ | 2048 | 68.60 |
| SimSiam (9) | ✗ | ✓ | ✗ | ✗ | 256 | 68.10 |
| INTL (53) | ✗ | ✗ | ✗ | ✗ | 1024 | 69.70 |
| MEC (32) | ✗ | ✓ | ✗ | ✗ | 1024 | 70.60 |
| Resa (52) | ✓ | ✗ | ✗ | ✓ | 256 | 70.80 |
| AdaDim (Ours) | ✓ | ✗ | ✗ | ✗ | 256 | 71.01 |
| Resa (52) | ✓ | ✓ | ✗ | ✓ | 256 | 71.90 |
| AdaDim (Ours) | ✓ | ✓ | ✗ | ✗ | 256 | 71.42 |

Table 4: This table shows a comparison in the solo-learn benchmark table with all methods using a ResNet-18 model trained for 1000 epochs on Cifar-10 and Cifar-100 with a batch size of 256. ImageNet-100 experiments are performed with 400 epochs and a batch size of 128. (M = Momentum Encoder, P = Predictor, Q = Queue, C = Clustering)

Table 5: This table shows a state of the art 100 epoch ImageNet comparison within the single crop setting. For AdaDim, we use the hyperparameters of our expanded setting. All comparison results are taken from their original papers or tables in (52).

dimension of 2048. The baseline setting also uses the LARS optimizer, batch size of 256, a learning rate of 0.4, temperature of 0.1, and a weight decay of 1e-4. We also use the asymmetric augmentation scheme of (11). Evaluation is performed with an online linear predictor that correlates with the offline setting (19; 18). In the expanded setting, we use the same AdaDim loss function, but integrate the architectural techniques of certain state of the art approaches such as momentum encoders. For these experiments, we use the training hyperparameters of the state of the art Resa algorithm (52) which uses the augmentation scheme of (59), an output projection size of 512, a SGD optimizer, batch size of 256, learning rate of 0.4, SGD momentum term of 0.9, temperature of 0.1, and weight decay of 1e-4. Evaluation is performed in the offline manner of (16). All experiments generally use 20 epochs between every $\alpha$ update and a default gamma of $1e - 4$. Further details are in Section A.4.

In Table 2, we analyze the performance impact of different hypothetical design choices on $\alpha$, $\gamma$, and $\beta$. Note that while AdaDim only introduces a single hyperparameter $\gamma$, we individually tune each in this table to validate the adaptive nature of our method. To start, we compare against methods that make use of heuristic $\alpha$ scaling methods without an adaptive computation such as cosine or linear schedules between 0 and 1 over the course of training. These methods underperform relative to the adaptive case and highlights the importance of adapting the optimization based on the dimensional characteristics of the dataset. We also compare against using a fixed $\gamma$ term during training. We observe that this regularization causes a slight decrease in performance. This result suggests that $I(R; Z)$ regularization should be applied selectively at specific points in SSL training, rather than a constant term throughout. This intuition is further confirmed by comparing against the $\lambda$ regularized $I(R; Z)$ reduction loss proposed in (35). This work argues that simply reducing $I(R; Z)$ during training without an adaptive mechanism can improve SSL representations. However, simply reducing $I(R; Z)$ doesn't conform with the dynamics we discuss where $I(R; Z)$ goes through periods of growth and reduction during training. Consequently, we find that using their suggested $\lambda$ parameter results in little to no improvement in our baseline setting. The reason for this discrepancy with their results may be that in their original paper their method only showed improvements with 200 epochs of training. In this limited setting, fixed regularization may work as there is limited training time for the discussed dynamics to emerge. We further validate these findings across more datasets and a fixed parameter setting in Section C.9. We also note a performance improvement when transitioning between training with $\alpha$ alone compared to $\beta$ in tandem with $\alpha$. This result confirms the importance of balancing between both $H(R)$ and $I(R; Z)$ regularization in a manner consistent with the discussed training dynamics. We confirm this ablation study across many different datasets in Table 3 and in comparison with state of the art approaches. We find that our method out performs or is comparable to other approaches across diverse settings with a fixed $\gamma$ choice of $1e - 4$. Additionally, further performance improvements can be attained by tuning $\gamma$ with respect to each dataset individually which is a result of better tuning towards the different dimensionality characteristics of each dataset. Furthermore, adding a momentum encoder as part of the AdaDim methodology leads to significant performance improvements for the majority of datasets. This emphasizes that AdaDim can effectively leverage the architectural techniques of other methods.

Since our method is motivated by adapting to training dynamics, we also analyze performance under different amounts of training epochs in part a) of Figure 6. We observe that our method out-performs competitive baselines regardless of training time. However, we also note that the margin of improvement is higher with more epochs which indicates that AdaDim better adapts to the dynamics of SSL methods. We further highlight this advantage in part b) of Figure 6 with 2000 epochs of training across 3 different datasets. Again, our method is the only one that consistently out-performs all others in the long training regime. We also visualize the $H(R)$ and $I(R; Z)$ trajectory across 2000 epochs for TinyImageNet in part c) of Figure 6. Our method does not have the highest $H(R)$ nor lowest $I(R; Z)$, but arrives at a balancing point between both.

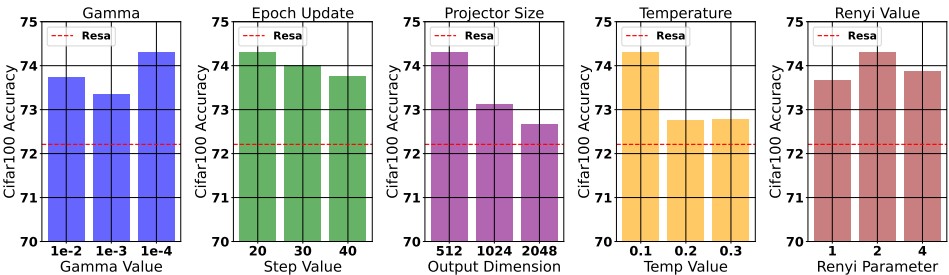

Figure 7: This figure shows the impact of different hyperparameter choices on performance with 1000 epochs of training on Cifar-100 with a ResNet-18 model. Experiments are performed in the expanded hyperparameter setting where AdaDim has an additional momentum encoder. For each plot, we indicate where the performance of the Resa (52) baseline lies in relation.

We also compare our method in the expanded hyperparameter setting in Tables 4 and 5 on standardized benchmarks (11; 13). We specify whether each comparison method utilizes additional SSL techniques such as a momentum encoder(M), predictor head (P), queue (Q), or clustering technique (C). In Table 4, we note that even without using any of these techniques, we are able to achieve performance that out-performs many of the other methods with the only additional overhead being an SVD calculation on 10 batches every 20 epochs. We validate the low compute cost of our method in Section A.8. However, we also show that our method makes use of a momentum encoder particularly well to out-perform all existing methods by a significant margin across all datasets. The reason for this improvement may be due to the momentum encoder acting as a stable update of the representation space thereby allowing AdaDim to better leverage the discussed training dynamics. We also observe similar improvements over baselines on the large scale ImageNet dataset where AdaDim is comparable to the most recent state of the art Resa algorithm (52) without requiring an additional clustering step. We also find that these improvements are consistent even with perturbations to the base set parameters in the expanded hyperparameter setting. In Figure 7, we alter a variety of parameters that could impact the performance of our method which includes the $\gamma$ value, the number of epochs between $\alpha$ updates, the output projector size, the temperature in $L_{NCE}$, and the renyi parameter applied when computing $L_{mut}$. In all cases, we maintain a performance improvement over the Resa method. Overall, this study demonstrates the experimental robustness of AdaDim.

## 6 CONCLUSION

This paper demonstrates theoretically and empirically that the best performing SSL models arrive at a balance between the dimensionality $H(R)$ of the representation space and the mutual information between the representation and embedding spaces $I(R; Z)$. Specifically, these dynamics indicate that increases in $H(R)$ due to feature decorrelation are preserved between $R$ and $Z$, but increases due to the samples spreading uniformly can cause $I(R; Z)$ to increase, plateau, or decrease depending on the stage of training of the SSL algorithm. We then introduce a method called AdaDim based on adapting $H(R)$ based on feature decorrelation and sample uniformity and gradual regularization of $I(R; Z)$. AdaDim results in improved performance over baseline SSL strategies without requiring additional architectural overhead. However, further performance improvements are possible when using an additional momentum encoder. This results in a significant margin of improvement over state of the art benchmarks.

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

# A    APPENDIX EXPERIMENTAL DETAILS

## A.1    CODEBASE

We use the solo-learn codebase (11) and the codebase associated with the Resa algorithm (52). Code for this paper will be released upon acceptance.

## A.2    DATASETS

We show explicit details of all datasets used in this paper in Table 6. The datasets were chosen to achieve as much diversity across a wide variety of data settings. This includes medical and natural image datasets, datasets of varying sizes, datasets of varying class complexity, and datasets with varying class imbalances.

| Dataset | Abbreviation & Link | Description | # of classes |
|---|---|---|---|
| CIFAR-100 (31) | cifar100 | 100 classes of 32x32 color images, including animals, vehicles, and various objects commonly found in the world. | 100 |
| CIFAR-10 (31) | cifar10 | 10 classes of 32x32 color images featuring everyday objects and scenes such as airplanes, cars, and animals. | 10 |
| TinyImageNet200 (55) | tinyimagenet200 | 200 classes of 64x64 images, a smaller version of the ImageNet dataset, used for object recognition and classification tasks. | 200 |
| BloodMNIST (54) | blood | 8 classes of 28x28 images, designed for classification of diseases in red blood cells. | 8 |
| OrganSMNIST (54) | organs | 11 classes of 28x28 images, designed for classifying various types of liver tumor problems. | 11 |
| OrganCMNIST (54) | organc | 11 classes of 28x28 images, designed for classifying various types of liver tumor problems. | 11 |
| OrganAMNIST (54) | organa | 11 classes of 28x28 images, designed for classifying various types of liver tumor problems. | 11 |
| STL10 (54) | stl10 | 10 classes of 96x96 images, designed for classifying various types of images. | 10 |
| Cinic-10 (12) | cinic10 | 10 classes of 96x96 images, designed for developing unsupervised feature learning, deep learning, and self-taught learning algorithms. | 10 |
| iNaturalist 2021 (49) | inat21 | Large-scale dataset with over 10,000 species, collected from photographs of plants and animals in their natural environments for fine-grained classification. | 10,000 |
| ImageNet (13) | imagenet | Large dataset with over 1,000 classes, used for image classification and object detection, containing millions of images across a wide variety of categories. | 1,000 |

Table 6: Overview of the datasets used in this paper.

## A.3 PSEUDO-CODE OF ADADIM

We note that AdaDim can be applied with an additional momentum encoder that is used in a variety of works (8; 52). In this setting, the momentum encoder is a separate network that is effectively a copy of the main backbone network. The difference is that a momentum encoder is not updated with backpropagation, but is instead updated as an exponential moving average of the weights of the main backbone network. During training, the encoder produces its own representation matrices that have previously been shown to aid in representational stability when integrated with other SSL methods. We can then perform the same AdaDim loss with the additional momentum encoding representations and embeddings as shown in the pseudocode above. Note that our projector has an output dimension of 512 and the final layer is passed through a 1D batch normalization step.

## A.4 METHOD TRAINING DETAILS

### A.4.1 BASELINE SETTING

We show the basic parameters for the baseline setting in Table 7 and the associated dataset specific parameters in Table 9. This setting uses the basic joint embedding architecture setup without any additional architectural techniques such as a momentum encoder or predictor head. In this setting, we also use the LARS optimizer, a 256 batch size, 1e-4 weight decay, 10 epochs of warmup, a $\gamma = 1e - 4$, a temperature of 0.1, projector output of 2048, and an $\alpha$ update every 20 epochs. The augmentation scheme is the asymmetric augmentation scheme of (11) described in Table 11.

For evaluation of our method, we use the online linear evaluation setting of (11) where the classifier is trained alongside the backbone and projector during SSL pre-training. Representations are fed to a linear classifier while keeping the gradient of the classifier's cross entropy loss from flowing through the backbone. The linear classifier has its own separate learning rate of 0.1 that follows a cosine annealing schedule based on the number of epochs of SSL training. The performance of the online classifier correlates well with the offline setting, making it a reliable proxy as shown in (19; 7; 11).

### A.4.2 EXPANDED SETTING

In the expanded setting, we use the same hyperparameters described in (52) who also borrow the parameters of (53). These parameters are used when we integrate a momentum encoder or predictor head on top of the standard joint embedding architecture that we use in the baseline setting. The basic parameters for this setting are described in Table 7 with dataset specific alterations described in Table 9. These parameters include a SGD optimizer, batch size of 256, base learning rate of 0.4, cosine scheduler, weight decay of 1e-4, 2 warmup epochs, $\gamma = 1e - 4$, temperature of 0.1, 512 output projector size and 20 epochs between every $\alpha$ update. The attached momentum encoder also has a momentum encoding update parameter of 0.996. This setting also uses the augmentation scheme of (59) described in Table 11.

Another difference in this setting is that evaluation is done in an offline manner where a linear layer is appended to the frozen SSL encoder and fine-tuned with specific hyperparameters. Specifically, we use the linear fine-tuning approach of (52) and (53) described in Table 8. For small scale experiments (i.e. datasets that are notImageNet), the linear layer is fine-tuned for 500 epochs, with an Adam optimizer, a learning rate that drops from 1e-2 to 1e-6, and a weight decay of 5e-6. Training is performed in a stochastic manner where subsets of the training set are used at each epoch. This setting allows evaluation to take place in a few minutes and is suitable for analyzing smaller datasets. For ImageNet and ImageNet-100 experiments, an SGD optimizer is used with a momentum parameter of 0.9, 100 epochs of training, a base learning rate and weight decay specified in Table 9, and step down scheduler by a factor of 10 at epochs 60 and 80.

### A.4.3 COMPARISON METHODS

All essential hyperparameters for comparisons with state of the art methods are shown in Table 10. Note that these parameters are used to train comparisons for the ablation study of Table 2 and the diverse data study of Table 3. For all other comparisons against past state of the art SSL methods, we use the benchmark tables provided in (11) for comparisons with Cifar-10, Cifar-100, and ImageNet-100. For comparisons with ImageNet, we copy the tables provided in (52) that is taken from the

**Algorithm 1** Pytorch style pseudocode for AdaDim

```
# E, Em: encoder, momentum encoder (optional)
# G, Gm: projector mlp, momentum projector mlp (optional)
# T1, T2: augmentation1, augmentation2
# temp: temperature = 0.1
# gama: gamma = 1e-4
# b: Batch size = 256
# vicreg_loss with default weightings of 25,25,1
# nce_loss
# alpha_renyi_loss with parameter of 2
# get_rank function computes effective rank
# alpha_epoch: epoch where the alpha parameter is computed
# current_epoch: Epoch of Current Training
import torch.nn.functional as F
###########################
# Loss without momentum encoder
###########################
for x in loader: # load a minibatch x with b samples
    x1, x2 = T1(x), T2(x) # two augmentation views
    r1, r2 = E(x1), E(x2) # representation space
    z1, z2 = G(h1), G(h2) # embeddings

    if alpha_epoch % current_epoch == 0:
        Z = torch.cat((z1,z2))
        min_dim = min(Z.shape[0],Z.shape[1])
        rank = get_rank(Z)
        alpha = rank / min_dim
        beta = alpha * gamma

    v_loss = vicreg_loss(z1,z2)
    z1_norm = F.normalize(z1)
    z2_norm = F.normalize(z2)
    n_loss = nce_loss(z1_norm,z2_norm,temperature)
    ssl_loss = alpha * v_loss + (1-alpha) * n_loss
    mut_loss = (alpha_renyi_loss(z1,r2) + alpha_renyi_loss(z2,r1)) / 2

    return (1-beta) * ssl_loss  - beta * mut_loss

###########################
# Loss with Momentum encoder
###########################
for x in loader: # load a minibatch x with b samples
    x1, x2 = T1(x), T2(x) # two augmentation views
    r1, r2 = E(x1), E(x2) # representation space
    z1, z2 = G(h1), G(h2) # embeddings

    with torch.no_grad():
        update_momentum_params(0.996 -> 1) # exponential moving average
        r1m, r2m = Em(x1), Em(x2) # momentum encodings
        z1m, z2m = Gm(h1m), Gm(h2m) # momentum embeddings

    if alpha_epoch % current_epoch == 0:
        Z = torch.cat((z1,z2m))
        min_dim = min(Z.shape[0],Z.shape[1])
        rank = get_rank(Z)
        alpha = rank / min_dim
        beta = alpha * gamma

    v_loss = (vicreg_loss(z1,z2m) + vicreg_loss(z1m,z2)) / 2
    z1_norm = F.normalize(z1)
    z2_norm = F.normalize(z2)
    z1m_norm = F.normalize(z1m)
    z2m_norm = F.normalize(z2m)
    n_loss = (nce_loss(z1_norm,z2m_norm,temperature) +
    ↪  nce_loss(z1m_norm,z2_norm,temperature)) / 2
    ssl_loss = alpha * v_loss + (1-alpha) * n_loss
    mut_loss = (alpha_renyi_loss(z1,r2) + alpha_renyi_loss(z2,r1))
    mut_loss_momentum = (alpha_renyi_loss(z1m,r2m) +
    ↪  alpha_renyi_loss(z2m,r1m))
    total_mut_loss = (mut_loss + mut_loss_momentum) / 4

    return (1-beta) * ssl_loss  - beta * total_mut_loss
```

| AdaDim Baseline Parameters | | | | | | | | | | | | |
|---|---|---|---|---|---|---|---|---|---|---|---|---|
| Method | Setting | M | P | Optimizer | Batch Size | Base LR | Weight Decay | Warmup | $\gamma$ | Temp | Alpha Update | Proj Output |
| AdaDim | Baseline | ✗ | ✗ | LARS | 256 | 0.4 | 1e-4 | 10 | 1e-4 | 0.1 | 20 | 2048 |
| AdaDim | Expanded | ✓ | ✓or✗ | SGD | 256 | 0.4 | 1e-4 | 2 | 1e-4 | 0.1 | 20 | 512 |

Table 7: This table shows the baseline parameters for most experiments. M refers to using a momentum encoder. P refers to using an additional prediction head.

| AdaDim Expanded Hyperparameter Specific Details | | | | | | | | | |
|---|---|---|---|---|---|---|---|---|---|
| Method | Setting | M | P | Pretraining Momentum | Linear Optimizer | Linear Base LR | Linear Weight Decay | Linear Scheduler | Linear Epochs |
| AdaDim | Expanded | ✓ | ✓or✗ | .996 | Adam | 0.01 | 5e-6 | Exponential | 500 |

Table 8: This table shows the momentum parameter used during SSL pre-training within a momentum encoder and the details of the offline linear evaluation setting for the momentum encoding experiments. M refers to using a momentum encoder. P refers to using an additional prediction head.

results reported in the original paper for each method. In the case of method specific hyperparameters, we use the parameters described in the solo-learn codebase as much as possible.

In Table 2, we also compare with the regularization technique of (35). This involves taking the matrices $R$ and $Z$ and computing the mutual information estimate based on the $\alpha$-Renyi approximation discussed in Section A.6. This regularization term is scaled by a $\lambda$ parameter that is set to 1e-4 for all experiments. This specific choice of $\lambda$ is based on the value that performed best in (35).

| AdaDim Dataset Specific Training | | | | | | | | | | | | |
|---|---|---|---|---|---|---|---|---|---|---|---|---|
| Method | Setting | Dataset | M | P | Tuned Gamma | Epochs | Model | Batch Size | Augmentation Scheme | Base Linear LR | Pretrain LR | WD |
| AdaDim | Baseline | Cifar100 | ✗ | ✗ | 1e-4 | 400 | ResNet-50 | 256 | Asymmetric | 0.1 | 0.4 | 1e-4 |
| AdaDim | Expanded | Cifar100 | ✓ | ✗ | 1e-4 | 400 | ResNet-50 | 256 | ReSSL | 0.01 | 0.4 | 1e-4 |
| AdaDim | Baseline | TinyImageNet200 | ✗ | ✗ | 1e-4 | 400 | ResNet-50 | 256 | Asymmetric | 0.1 | 0.4 | 1e-4 |
| AdaDim | Expanded | TinyImageNet200 | ✓ | ✗ | 1e-4 | 400 | ResNet-50 | 256 | ReSSL | 0.01 | 0.4 | 1e-4 |
| AdaDim | Baseline | Cinic10 | ✗ | ✗ | 1e-2 | 400 | ResNet-50 | 256 | Asymmetric | 0.1 | 0.4 | 1e-4 |
| AdaDim | Expanded | Cinic10 | ✓ | ✗ | 1e-4 | 400 | ResNet-50 | 256 | ReSSL | 0.01 | 0.4 | 1e-4 |
| AdaDim | Baseline | STL10 | ✗ | ✗ | 1e-2 | 400 | ResNet-50 | 256 | Asymmetric | 0.1 | 0.4 | 1e-4 |
| AdaDim | Expanded | STL10 | ✓ | ✗ | 1e-4 | 400 | ResNet-50 | 256 | ReSSL | 0.01 | 0.4 | 1e-4 |
| AdaDim | Baseline | BloodMNIST | ✗ | ✗ | 1e-4 | 400 | ResNet-50 | 256 | Asymmetric | 0.1 | 0.4 | 1e-4 |
| AdaDim | Expanded | BloodMNIST | ✓ | ✗ | 1e-4 | 400 | ResNet-50 | 256 | ReSSL | 0.01 | 0.4 | 1e-4 |
| AdaDim | Baseline | OrganSMNIST | ✗ | ✗ | 1e-4 | 400 | ResNet-50 | 256 | Asymmetric | 0.1 | 0.4 | 1e-4 |
| AdaDim | Expanded | OrganSMNIST | ✓ | ✗ | 1e-4 | 400 | ResNet-50 | 256 | ReSSL | 0.01 | 0.4 | 1e-4 |
| AdaDim | Baseline | OrganCMNIST | ✗ | ✗ | 1e-2 | 400 | ResNet-50 | 256 | Asymmetric | 0.1 | 0.4 | 1e-4 |
| AdaDim | Expanded | OrganCMNIST | ✓ | ✗ | 1e-4 | 400 | ResNet-50 | 256 | ReSSL | 0.01 | 0.4 | 1e-4 |
| AdaDim | Baseline | OrganAMNIST | ✗ | ✗ | 1e-1 | 400 | ResNet-50 | 256 | Asymmetric | 0.1 | 0.4 | 1e-4 |
| AdaDim | Expanded | OrganAMNIST | ✓ | ✗ | 1e-4 | 400 | ResNet-50 | 256 | ReSSL | 0.01 | 0.4 | 1e-4 |
| AdaDim | Baseline | Cifar100 | ✗ | ✗ | 1e-4 | 1000 | ResNet-18 | 256 | Asymmetric | 0.1 | 0.4 | 1e-4 |
| AdaDim | Baseline | Cifar10 | ✗ | ✗ | 1e-4 | 1000 | ResNet-18 | 256 | Asymmetric | 0.1 | 0.4 | 1e-4 |
| AdaDim | Baseline | ImageNet100 | ✗ | ✗ | -1e-1 | 400 | ResNet-18 | 256 | Asymmetric | 0.1 | 0.4 | 1e-4 |
| AdaDim | Expanded | Cifar100 | ✓ | ✗ | 1e-4 | 1000 | ResNet-18 | 256 | ReSSL | 0.01 | 0.4 | 1e-4 |
| AdaDim | Expanded | Cifar10 | ✓ | ✗ | 1e-4 | 1000 | ResNet-18 | 256 | ReSSL | 0.01 | 0.4 | 1e-4 |
| AdaDim | Expanded | ImageNet100 | ✓ | ✗ | 1e-4 | 400 | ResNet-18 | 128 | ReSSL | 5 | 0.5 | 2.5e-5 |
| AdaDim | Expanded | ImageNet | ✓ | ✗ | 1e-4 | 100 | ResNet50 | 256 | ReSSL | 2 | 0.5 | 1e-5 |
| AdaDim | Expanded | ImageNet | ✓ | ✓ | 1e-4 | 100 | ResNet50 | 256 | ReSSL | 1 | 0.5 | 1e-5 |

Table 9: This table shows the details of specific choices made on a per dataset basis for the tables generated in the main paper. M refers to using a momentum encoder. P refers to using an additional prediction head.

## A.5 COMPLETE SimCLR AND VICReg Loss

In this section, we go into more depth regarding the $L_{NCE}$ and $L_{VICReg}$ losses. Suppose there is an image $i$ drawn from a training pool $i \in I$. $i$ is passed into two random transformations $t(i) = x$ and $t^{'}(i) = x^{'}$ where $t$ and $t^{'}$ are drawn from the set of all random augmentations $T$. Both $x$ and $x^{'}$ are passed into an encoder network $e(\cdot)$. This results in the representations $e(x) = r$ and $e(x^{'}) = r^{'}$. These representations are then passed into a projection head $g(\cdot)$ that produces the embeddings $g(x) = z$ and $g(x^{'}) = z^{'}$. The collection of all representations and embeddings within a batch of $n$ samples can be represented by the $R$, $R^{'}$, $Z$, and $Z^{'}$ matrices. In this case, all matrices are composed of $n$ vectors with dimension $D$. This can be written as $R = [r_1, r_2, ..., r_n]$, $R^{'} = [r^{'}_1, r^{'}_2, ..., r^{'}_n]$, $Z^{'} = [z^{'}_1, z^{'}_2, ..., z^{'}_n]$, and $Z = [z_1, z_2, ..., z_n]$. From this setup, the VICReg (3) and InfoNCE (34; 7) losses can be computed. In this case, VICReg corresponds to a feature decorrelation loss that is better at promoting higher $H(R)$ while InfoNCE corresponds to a sample uniformity loss better at promoting lower $I(R; Z)$ at the end of training.

| Comparison Method Parameters | | | | | | | | | |
|---|---|---|---|---|---|---|---|---|---|
| Method | Setting | Specific Parameters | Model | Epochs | Optimizer | Learning Rate | Batch Size | Augmentation Scheme | Projector Output |
| SimCLR | Baseline | temp = 0.1 | ResNet-50 | 400 | LARS | 0.4 | 256 | Symmetric | 128 |
| VICReg | Baseline | Params= 25,25,1 | ResNet-50 | 400 | LARS | 0.4 | 256 | Asymmetric | 2048 |
| Moco v2 | Baseline | temp = 0.2, momentum = [0.9,0.99], queue = 32768 | ResNet-50 | 400 | SGD | 0.3 | 256 | weak_symmetric | 256 |
| BYOL | Baseline | momentum = 1.0, base = 0.99 | ResNet-50 | 400 | LARS | 1.0 | 256 | asymmetric | 256 |
| Barlow Twins | Baseline | scale_loss = 0.1 | ResNet-50 | 400 | LARS | 0.3 | 256 | asymmetric | 2048 |
| NNCLR | Baseline | queue = 65536, temp = 0.2 | ResNet-50 | 400 | LARS | 0.4 | 256 | asymmetric | 256 |
| SimSiam | Baseline | temp = 0.2 | ResNet-50 | 400 | LARS | 0.5 | 256 | weak_symmetric | 512 |
| DeepCluster v2 | Baseline | prototypes = [3000, 3000, 3000] | ResNet-50 | 400 | LARS | 0.6 | 256 | symmetric | 128 |
| Moco v3 | Baseline | momentum = [0.9, 0.99] | ResNet-50 | 400 | LARS | 0.3 | 256 | asymmetric | 256 |

Table 10: This table shows the training details of the models we trained directly for the diverse dataset study in the main paper. The same hyperparameters were maintained across all datasets.

| Augmentation Details | | | | |
|---|---|---|---|---|
| | Asymmetric Augmentation | | ReSSL Augmentation | |
| Augmentation Type | View 1 | View 2 | View 1 | View 2 |
| Crop | 1 | 1 | 1 | 1 |
| Color Jitter | 0.8 | 0.8 | 0.0 | 0.8 |
| Contrast | 0.4 | 0.4 | 0.0 | 0.4 |
| Brightness | 0.4 | 0.4 | 0.0 | 0.4 |
| Saturation | 0.2 | 0.2 | 0.0 | 0.2 |
| Hue | 0.1 | 0.1 | 0.0 | 0.1 |
| Grayscale | 0.2 | 0.2 | 0.0 | 0.2 |
| Gaussian Blur | 1.0 | 0.1 | 0.0 | 0.0 |
| Solarization | 0.0 | 0.2 | 0.0 | 0.2 |
| Equalization | 0.0 | 0.0 | 0.0 | 0.0 |
| Horizontal Flip | 0.5 | 0.5 | 0.5 | 0.5 |

Table 11: In this table, we detail the augmentations applied to each view (View 1 and 2) during pre-training of our algorithm. We divide the augmentations based on the type of scheme we are using. Every value in the table represents the probability of a specific augmentation type being applied.

The InfoNCE ($L_{NCE}$) loss is written as: $L_{NCE} = -\sum_{i \in I} log \frac{exp(sim(z_i, z_i')/\tau)}{\sum_{k=1}^{2N} \mathbb{1}[k \neq i] exp(sim(z_i, z_k))}$ where $sim$ refers to the cosine similarity, $\tau$ represents a temperature parameter, and the summation in the denominator takes place over all samples from both transformations. The VICReg loss is written as: $L_{VICReg} = \lambda s(Z, Z') + \mu[v(Z) + v(Z')] + \nu[c(Z) + c(Z')]]$. The invariance term is $s(Z, Z') = \frac{1}{n} \sum_{i=1}^{N} ||z_i - z_i'||_2^2$. The covariance term is $c(Z) = \frac{1}{D} \sum_{i \neq j} [C(Z)]_{i,j}^2$ where $C(Z)$ is the covariance matrix of $Z$. The variance term is $v(Z) = \frac{1}{d} \sum_{j=1}^{D} max(0, \gamma - S(z^j, \epsilon))$ where $S(x, \epsilon)$ is the regularized standard deviation, $z^j$ represents the vector of each value at dimension $j$, and $\gamma$ is a target value set to 1 for all experiments. For both $L_{NCE}$ and $L_{VICReg}$, we use the conventions of the original papers which includes $\tau = 0.1$, $\lambda = \mu = 25$, and $\nu = 1$.

## A.6 METRIC ANALYSIS DETAILS

One possible mathematical description for the dimensionality of a representation space $H(R)$ is the von Neumann entropy of eigenvalues (50; 25) which takes the form $H(R) = -\sum_i \lambda_i log(\lambda_i)$ where each $\lambda_i$ represents an eigenvalue of $R$. To increase $H(R)$ in this formula, we can either increase the total number of non-zero eigenvalues or maintain the same number of eigenvalues, but make the eigenvalues more similar in value to each other (higher uniformity, lower variance). Increasing the total number of eigenvalues corresponds to feature decorrelation in which an SSL algorithm discovers a larger number of total dimensions along which $R$ can vary. Decreasing the variance of eigenvalues within a fixed dimensional space corresponds to sample uniformity where representations spread more equally along all dimensions.

Throughout the paper, the dynamics between $H(R)$ and $I(R; Z)$ is discussed. However, this analysis requires a variety of metrics that were not fully detailed in the main paper. For our analytical experiments, the test set of interest is passed into the trained SSL model and its associated projection head. This results in a matrix for the representation space $R$ and a corresponding matrix for the embedding space $Z$. These matrices are of size: number of test set samples $\times$ 2048. These matrices

are then used to compute the metrics used for analysis in the main paper. $I(R; Z)$ is computed using the $\alpha$-Renyi matrix mutual information approximation discussed in (44). To calculate this quantity, assume that normalized matrices A and B are both $R^{nxn}$. The entropy of matrix A can be represented as $H_\alpha(A) = \frac{1}{1-\alpha} log[tr((\frac{A}{n})^\alpha)]$ where $\alpha = 2$ for all experiments. This formulation results in a matrix mutual information estimator of the form $I(A; B) = H_\alpha(A) + H_\alpha(B) - H_\alpha(A \odot B)$ where $\odot$ is the hadamard product. This formulation only works for positive semi definite matrices so during our experiments the approximation of (44) is followed where the normalized covariance matrices $RR^T$ and $ZZ^T$ are used as inputs to calculate $I(R; Z)$.

Note that there are a variety of ways to approximate $H(R)$. In this paper, both $H_\alpha(R)$ and the effective rank(38) are used at different points. The main reason for this choice is that the effective rank is normalized with respect to the eigenvalues of the current distribution. This means that the lowest possible value is 0 and the highest possible value is minimum dimension of the matrix of interest. The advantage of the $\alpha$-Renyi approximator is that the scale of the values will more closely match the values used to calculate $I(R; Z)$. However, both metrics result in the same balancing behavior between $H(R)$ and $I(R; Z)$ and are correlated with each other. This correlation is observed in Figure 8. In general, any computation of $H(R)$ can be thought of as an approximation of the dimensionality of the representation space. This is because higher dimensionality has been characterized in terms of eigenvalue distributions across a variety of works (19; 45; 2; 22). These metrics follow this trend as they are based on measuring how closely the eigenvalue spectrum of a given matrix approaches a uniform distribution. For example, another possible entropy estimator is discussed in (58). This work states that for a positive semi definite (PSD) matrix $A$, matrix entropy (ME) can be defined as $ME(A) = -tr(Alog(A)) + tr(A) = -\sum_i \lambda_i log(\lambda_i) + \sum_i \lambda_i$. The first term will increase with the sample uniformity of the representation space i.e. as the eigenvalues become more uniformly distributed. The second term will increase with more and larger eigenvalues i.e. as the features of the space become more decorrelated.

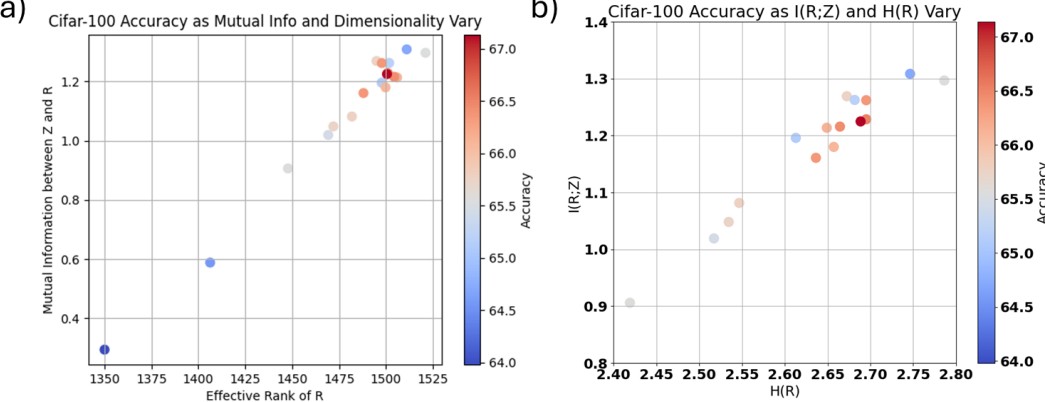

Figure 8: We show versions of the same opening Figure with $H(R)$ computed with a) the effective rank and b) an $\alpha$-Renyi matrix approximator.

The uniformity metric (51) is also used as part of our analysis. This metric acts as a measurement of how uniformly distributed the points of a representation space are on a hypersphere. It takes the form of the pairwise gaussian potential kernel and can be expressed as $log(\mathbb{E}_{(x,y) \sim p_{data}}[e^{-2||e(x)-e(y)||_2^2}])$. In general, greater uniformity indicates a more negative value when this metric is computed empirically. We use the implementation from the original github of (51).

## A.7 COMPUTE RESOURCES

Our resources included a personal PC with 8 Intel i7-6700K CPU Cores and 2 12 GB Nvidia GeForce GTX Titan X GPUs. We also used a lab work station server with 12 Intel i7-5930K CPU cores and 2 24GB Nvidia TITAN RTX 3090 GPUs. We also used a server with compute resources based on availability and priority queues. The vast majority of experiments use these resources and are found

| Compute Analysis of 100 Epochs of Cifar-100 Training | | | |
|---|---|---|---|
| | SimCLR | VICReg | AdaDim | AdaDim + momentum |
| s/epoch | 22.01 | 19.09 | 25.48 | 30.82 |

Table 12: This table shows the compute cost of 100 epochs of training of Cifar-100 with a ResNet-18 model. We compute the total time for the experiment and divide by the number of epochs to get the seconds/epoch metric (s/epoch). All experiments are performed in identical settings with the same number of CPU cores and the usage of an RTX 3090 GPU. For these experiments, the same data augmentation strategy was applied for ease of comparability.

in the main paper or the appendix. However, there may be early exploratory experiments in the development of our method that were not included.

### A.8 COMPUTE DISCUSSION OF OUR METHOD

Our method involves computing the eigenspectrum at different points in the training process. In general, computing eigenvalues is an expensive operation with order $O(n^3)$. However, the number of calculations is limited through a few mechanisms specific to `AdaDim`. This includes the usage of the $E_\alpha$ parameter. This parameter dictates how many epochs must pass before the $\alpha$ parameter is re computed. In Figure 7, performance improvements are maintained even when $E_\alpha$ is varied. Additionally, for every $E_\alpha$, eigenvalues are computed for only 10 training batches. This design choice is based on the empirical observation that most batches have a similar effective rank as training progresses. This limits the need to compute the eigenvalues across all batches in an epoch. The averaging across 10 batches is done to ensure additional stability regarding the $\alpha$ update. However, it may be possible to use even fewer batches in this computation.

To further analyze the compute cost of our method, we perform an experiment where we measure the amount of time it takes to train a ResNet-18 model for 100 epochs on Cifar-100 with the same compute resources across all methods of interest. This setup is similar to the compute analysis of other papers such as (52). The results of this experiment are shown in Table 12. We observe that AdaDim does cost more than the baseline SimCLR or VICReg methods. This cost requirement increases further when adding an additional momentum encoder. However, the cost increase is not so drastic to make training with AdaDim a prohibitive endeavor. We observe minor increases in compute cost of 3 - 8 s/epoch above the baseline SimCLR approach, depending on whether we use the baseline AdaDim strategy or an additional momentum encoder network.

### A.9 EMPIRICAL EIGENVALUE ANALYSIS DETAILS

In Figure 3, we perform a variety of analyses on the eigenvalue distribution of the output matrices of a ResNet-50 model. In part b), all eigenvalues are normalized before counting the number of eigenvalues above a threshold $\tau$ that we set to .01 for all experiments. This normalization is done by dividing all eigenvalues by the l-1 norm of the total eigenvalue spectrum. This is similar to the normalization done in the computation of the effective rank. In part c), the cumulative explained variance ratio metric is computed. To compute this metric, assume that there is a set of eigenvalues $\lambda = [\lambda_1, \lambda_2, ..., \lambda_N]$ where the eigenvalues are ordered from largest to smallest. Assume that there is a percentage $p$ of the largest eigenvalues. This results in the explained variance metric: $\frac{\sum_{i=1}^{p*N} \lambda_i}{\sum_{i=1}^{N} \lambda_i}$. This metric increases as the subset of eigenvalues that we sum over constitutes more of the overall variance of the data. However, it will decrease as the spread of this variance is distributed over eigenvalues outside of the percentage that the numerator is summed over.

### A.10 RANDOM PARAMETER ABLATION STUDY

In Figure 4, we show how accuracy varies as a function of $H(R)$ and $I(R; Z)$ for a variety of models with different hyperparameters. We also compute the correlation of these results with metrics of interest in Table 1. We generate 15 models for 3 different methods on Cifar-100 and display the exact parameters for each of these methods in Table 13.

| Method | Dataset | Epochs | Parameters | Learning Rate | Temperature | Weight Decay | Effective Rank | Mutual Info | Accuracy |
|--------|---------|--------|-----------|--------------|-------------|--------------|----------------|-------------|----------|
| SimCLR | Cifar-100 | 100 | d=2048 | 0.6 | 0.05 | 10-6 | 673 | 0.073 | 47.15 |
| SimCLR | Cifar-100 | 100 | d=2048 | 0.6 | 0.07 | 10-6 | 682 | 0.071 | 48.84 |
| SimCLR | Cifar-100 | 100 | d=2048 | 0.6 | 0.1 | 10-6 | 684 | 0.079 | 49.43 |
| SimCLR | Cifar-100 | 100 | d=2048 | 0.6 | 0.2 | 10-6 | 646 | 0.075 | 52.28 |
| SimCLR | Cifar-100 | 100 | d=2048 | 0.6 | 0.3 | 10-6 | 594 | 0.076 | 54.39 |
| SimCLR | Cifar-100 | 100 | d=2048 | 0.6 | 0.4 | 10-6 | 566 | 0.068 | 51.99 |
| SimCLR | Cifar-100 | 100 | d=2048 | 0.5 | 0.05 | 10-6 | 658 | 0.064 | 48.65 |
| SimCLR | Cifar-100 | 100 | d=2048 | 0.5 | 0.07 | 10-6 | 652 | 0.056 | 47.36 |
| SimCLR | Cifar-100 | 100 | d=2048 | 0.5 | 0.1 | 10-6 | 670 | 0.055 | 54.56 |
| SimCLR | Cifar-100 | 100 | d=2048 | 0.5 | 0.15 | 10-6 | 666 | 0.074 | 54.03 |
| SimCLR | Cifar-100 | 100 | d=2048 | 0.5 | 0.2 | 10-6 | 637 | 0.078 | 54.98 |
| SimCLR | Cifar-100 | 100 | d=2048 | 0.5 | 0.3 | 10-6 | 587 | 0.078 | 54.15 |
| SimCLR | Cifar-100 | 100 | d=2048 | 0.5 | 0.4 | 10-6 | 556 | 0.07 | 53.1 |
| SimCLR | Cifar-100 | 100 | d=2048 | 0.5 | 0.15 | 10-7 | 655 | 0.076 | 54.86 |
| SimCLR | Cifar-100 | 100 | d=2048 | 0.5 | 0.15 | 10-5 | 666.67 | 0.076 | 53.59 |
| VICReg | Cifar-100 | 100 | nu = 0.3 | 0.3 | N/A | 10-6 | 922 | 0.268 | 50.75 |
| VICReg | Cifar-100 | 100 | nu = 0.4 | 0.3 | N/A | 10-6 | 914 | 0.265 | 50.83 |
| VICReg | Cifar-100 | 100 | nu = 0.5 | 0.3 | N/A | 10-6 | 902 | 0.292 | 50.62 |
| VICReg | Cifar-100 | 100 | nu = 0.6 | 0.3 | N/A | 10-6 | 892 | 0.263 | 52.29 |
| VICReg | Cifar-100 | 100 | nu = 0.7 | 0.3 | N/A | 10-6 | 902 | 0.235 | 56.72 |
| VICReg | Cifar-100 | 100 | nu = 0.8 | 0.3 | N/A | 10-6 | 903 | 0.24 | 56.27 |
| VICReg | Cifar-100 | 100 | nu = 0.9 | 0.3 | N/A | 10-6 | 898 | 0.237 | 55.73 |
| VICReg | Cifar-100 | 100 | nu = 1.0 | 0.3 | N/A | 10-6 | 883 | 0.244 | 52.37 |
| VICReg | Cifar-100 | 100 | nu = 1.1 | 0.3 | N/A | 10-6 | 878 | 0.229 | 57.6 |
| VICReg | Cifar-100 | 100 | nu = 1.2 | 0.3 | N/A | 10-6 | 877 | 0.232 | 52.28 |
| VICReg | Cifar-100 | 100 | nu = 1.3 | 0.3 | N/A | 10-6 | 847 | 0.257 | 54.54 |
| VICReg | Cifar-100 | 100 | nu = 1.4 | 0.3 | N/A | 10-6 | 868 | 0.212 | 55.09 |
| VICReg | Cifar-100 | 100 | nu = 1.5 | 0.3 | N/A | 10-6 | 795 | 0.279 | 49.19 |
| VICReg | Cifar-100 | 100 | nu = 1.6 | 0.3 | N/A | 10-6 | 867 | 0.209 | 54.48 |
| VICReg | Cifar-100 | 100 | nu = 1.7 | 0.3 | N/A | 10-6 | 848 | 0.2107 | 58.69 |
| NNCLR | Cifar-100 | 100 | d=2048 | 0.6 | 0.05 | 10-6 | 416 | 0.043 | 54.09 |
| NNCLR | Cifar-100 | 100 | d=2048 | 0.6 | 0.07 | 10-6 | 417 | 0.038 | 54.27 |
| NNCLR | Cifar-100 | 100 | d=2048 | 0.6 | 0.1 | 10-6 | 459 | 0.033 | 55.47 |
| NNCLR | Cifar-100 | 100 | d=2048 | 0.6 | 0.2 | 10-6 | 493 | 0.058 | 55.9 |
| NNCLR | Cifar-100 | 100 | d=2048 | 0.6 | 0.3 | 10-6 | 490 | 0.067 | 54.43 |
| NNCLR | Cifar-100 | 100 | d=2048 | 0.6 | 0.4 | 10-6 | 519 | 0.067 | 55.07 |
| NNCLR | Cifar-100 | 100 | d=2048 | 0.5 | 0.05 | 10-6 | 425 | 0.042 | 54.59 |
| NNCLR | Cifar-100 | 100 | d=2048 | 0.5 | 0.07 | 10-6 | 439 | 0.036 | 55.16 |
| NNCLR | Cifar-100 | 100 | d=2048 | 0.5 | 0.1 | 10-6 | 462 | 0.033 | 56.01 |
| NNCLR | Cifar-100 | 100 | d=2048 | 0.5 | 0.15 | 10-6 | 474 | 0.05 | 56.09 |
| NNCLR | Cifar-100 | 100 | d=2048 | 0.5 | 0.2 | 10-6 | 505 | 0.06 | 56.37 |
| NNCLR | Cifar-100 | 100 | d=2048 | 0.5 | 0.3 | 10-6 | 518 | 0.074 | 55.02 |
| NNCLR | Cifar-100 | 100 | d=2048 | 0.5 | 0.4 | 10-6 | 520 | 0.075 | 53.43 |
| NNCLR | Cifar-100 | 100 | d=2048 | 0.5 | 0.15 | 10-7 | 492 | 0.048 | 56.19 |
| NNCLR | Cifar-100 | 100 | d=2048 | 0.5 | 0.15 | 10-6 | 474 | 0.051 | 56.09 |

Table 13: This table shows all the parameters, accuracies, rank scores, and mutual information values for the random parameter experiments shown in the main paper.

## B    Appendix Theoretical Details

### B.1    High Level Intuition

Higher dimensionality in $R$ is desirable because it counters the dimensional collapse effect discussed in (22) and encourages a more diverse feature space. Lower $I(R; Z)$ is also desirable because it implies that the projection head is effective in removing uninformative features from the representation space. However, we prove through information theoretic bounds that increasing the dimensionality of $R$ causes a corresponding increase in $I(R; Z)$ thus necessitating a balance between the two for an ideal representation space. This balancing act is illustrated in Figure 1 where an image is passed through an encoder $e(\cdot)$ to produce a representation space $R$ with 6 associated features. 3 features are target-relevant and 3 are uninformative. The feature space is associated with an eigenvalue distribution that indicates how relevant each feature is to the geometry of the representation space. Ideally, the eigenvalue distribution should capture just the target-relevant features; however, a higher dimensional space also captures uninformative features as shown in part a). To counter this, the projector should act as an information bottleneck (47) during training that projects the features into a lower dimensional space where only the target features are relevant. In part a), the distribution of eigenvalues remains the same after projection so the projection head does not remove spurious features from $R$ which corresponds to a high $I(R; Z)$. Part b) represents an ideal case where $R$ has sufficiently high dimensionality to capture mostly informative features while sufficiently low $I(R; Z)$ such that the projector guides the optimization towards target-relevant features.

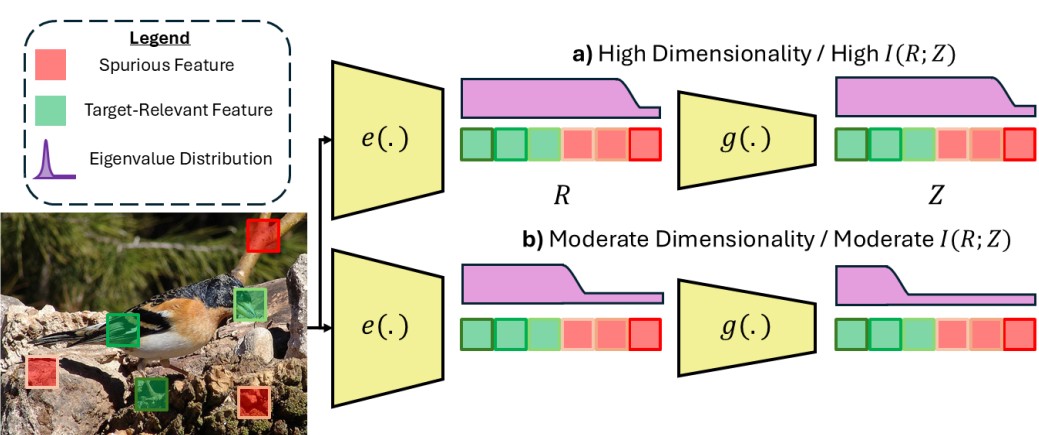

Figure 9: Assume there is an image with 3 task relevant features and 3 spurious features. The image is associated with a representation space $R$, projection space $Z$, and corresponding eigenvalue distributions for both. a) This is an example of $R$ and $Z$ with high dimensionality and high $I(R; Z)$. b) This is an example of $R$ and $Z$ that has moderate dimensionality and moderate $I(R; Z)$.

### B.2    Gaussian Mutual Info Derivation Details

We follow the assumptions of Section 3. The following closed form equations are needed for this analysis:

$$I(R; Z) = \frac{1}{2}(ln(|\Sigma_R|) + ln(|\Sigma_Z|) - ln(|\Sigma|))$$

$$H(R) = \frac{m}{2}ln(2\pi) + \frac{1}{2}ln(|\Sigma_R|) + \frac{m}{2}$$

$$ln(|\Sigma|) = ln(|\Sigma_Z||\Sigma_R - \Sigma_{RZ}\Sigma_Z^{-1}\Sigma_{ZR}|)$$

$$ln(|\Sigma|) = ln(|\Sigma_R||\Sigma_Z - \Sigma_{ZR}\Sigma_R^{-1}\Sigma_{RZ}|)$$

Note that the $ln(|\Sigma|)$ derivation originated from Shur's complement formula.

In this setting, $I(R; Z)$ can be rewritten as:

$$I(R; Z) = \frac{1}{2}(ln(|\Sigma_R|) + ln(|\Sigma_Z|) - ln(|\Sigma_R||\Sigma_Z - \Sigma_{ZR}\Sigma_R^{-1}\Sigma_{RZ}|))$$

$$I(R; Z) = \frac{1}{2}(ln(|\Sigma_R|) + ln(|\Sigma_Z|) - ln(|\Sigma_Z||\Sigma_R - \Sigma_{RZ}\Sigma_Z^{-1}\Sigma_{ZR}|))$$

Using the law of logarithms, we can simplify this equation into:

$$I(R; Z) = \frac{1}{2}(ln(|\Sigma_Z|) - ln(|\Sigma_Z - \Sigma_{ZR}\Sigma_R^{-1}\Sigma_{RZ}|))$$

$$I(R; Z) = \frac{1}{2}(ln(|\Sigma_R|) - ln(|\Sigma_R - \Sigma_{RZ}\Sigma_Z^{-1}\Sigma_{ZR}|))$$

This results in the form described in the main paper as:

$$I(R; Z) = \frac{1}{2}(ln(|\Sigma_Z|) - ln(|Var(Z|R)|)) = \frac{1}{2}(ln(|\Sigma_R|) - ln(|Var(R|Z)|))$$

We further analyze the specific terms that make up this equation in Figure 10. In parts a) and b), the $I(R; Z)$ curves from the main paper are repeated. In part c), each of the terms that make up $I(R; Z)$ are analyzed the number of features are fixed and the sample variance is increased. $ln(|\Sigma_R|)$ and $ln(|Var(Z|R)|)$ increases as the variance increases. However, $ln(|Var(Z|R)|)$ increases at a comparatively faster rate. This happens because $ln(|\Sigma_Z|)$ does not change in value. The end result is a reduction in mutual information which shows that $Z$ is not able to preserve the variance in $R$ under the conditions of its projection.

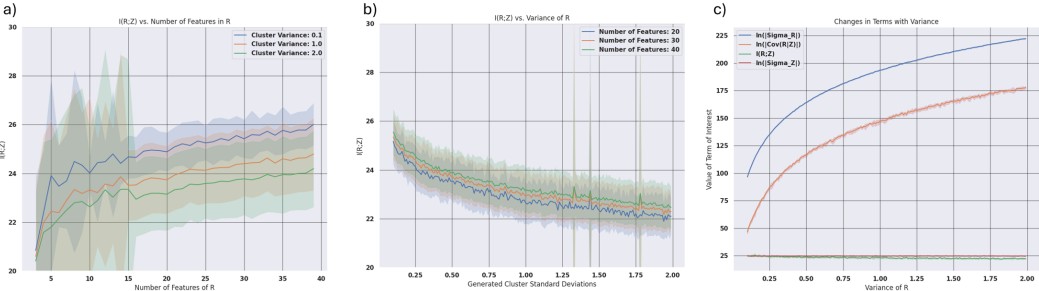

Figure 10: In a) and b), we show the $I(R; Z)$ curves as the number of features is varied while the cluster variance is kept constant. In c), we decompose each of the individual terms that make up $I(R; Z)$ and plot how their values vary during this same simulation .

### B.3 GAUSSIAN SIMULATION DETAILS

In Figure 2, we perform a detailed simulation on data generated from a Gaussian distribution. To generate this data, the make blobs dataset from the sklearn library(36) is used. This library generates Gaussian isotropic clusters that are intended for clustering problems. However, for our purposes it acts as a reliable generator of Gaussian distributed data. The cluster labels of this dataset are not used in any capacity for our experiments to conform to the SSL setting. This dataset has the following parameters:

1. n_samples: We set this to 1000 for all experiments.

2. n_features: We set this based on the features required for the simulation of interest.

3. centers: This is set to 5 for all experiments. This describes the number of clusters generated.

4. cluster_std: This is the parameter we vary to control the variance of the generated data.

5. random_state: This can set the initial random seed for the generation. We do not set this parameter. This is an intentional design choice so that we generate a slightly different version of the dataset on every iteration of the simulation. We take the average and standard deviation of 100 simulations for every set of parameters that we use in our experiments.

## B.4 NEURAL NETWORK SIMULATION

In the main paper, PCA is used as a general projection between $R$ and $Z$ for the purposes of modeling the interaction between a space and its projection without having to deal with the nuances of training neural networks. However, the projector can also be replaced with a neural network and trained with either the NCE (SimCLR) or VICReg loss. We show that even in this setting the same general trends hold.

For this experiment, synthetic gaussian data is generated in the manner described in Section B.3. For training, a small MLP is used composed of 5 layers and 20 hidden units per layer, followed by a small projector with 2 layers and 5 hidden units per layer to output a dimension of size 5. The generated data has 25 features and a cluster standard deviation of .01. It is trained for 1000 epochs with the NCE (SimCLR) or VICReg loss. In this setting, augmentations are generated by adding randomly distributed Gaussian noise with a standard deviation of 0.5 to the generated data. During training, $I(R; Z)$ is measured for every epoch where $R$ is the original generated data and $Z$ is the output of the neural network. This value is computed using the closed form $I(R; Z)$ for gaussian distributed data. The Adam optimizer is used for these experiments with a learning rate of .0001 and a $\beta$ of 0.9 to 0.999.

Figure 11 shows that the neural network simulation of our data exhibits the same trends as the PCA experiments found in the main paper. At the start of training, $I(R; Z)$ increases and gradually plateaus by the end of training. Additionally, the dimension contrastive strategy VICReg approaches a higher $I(R; Z)$ than that of the sample contrastive strategy SimCLR.

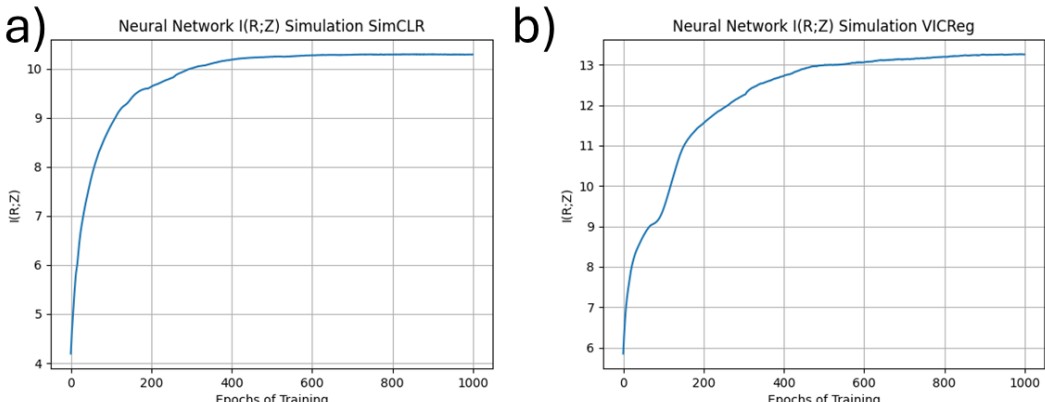

Figure 11: We show the $I(R; Z)$ curves across epochs of training for a gaussian dataset trained on a) SimCLR and b) VICReg.

## B.5 INFO THEORETIC BOUNDS

The upper bound on $I(R; Y)$ described in the main text originated from a derivation performed in (35). The exact details of these bounds can be found in the original paper. Below we show a complete derivation of the bound described in equation 3. The original bound is described as:

$$I(Y;R) \leq I(Y;Z) - I(R;Z) + H(R)$$

The approximation $I(Y;Z) = G$ is used in the main paper which results in the following bound:

$$I(Y;R) \leq G - I(R;Z) + H(R)$$

We substitute in the equation $\frac{1}{2}(ln(|\Sigma_Z|) - ln(|Var(Z|R)|))$ for $I(R;Z)$ and $H(R) = \frac{m}{2}ln(2\pi) + \frac{1}{2}ln(|\Sigma_R|) + \frac{m}{2}$. This results in the bound:

$$I(Y;R) \leq G - \frac{1}{2}(ln(|\Sigma_Z|) - ln(|Var(Z|R)|)) + (\frac{m}{2}ln(2\pi) + \frac{1}{2}ln(|\Sigma_R|) + \frac{m}{2})$$

A simplification of terms results in the bound shown in the main paper as:

$$I(Y;R) \leq G + \underbrace{\frac{1}{2}(ln(|\Sigma_R|) - ln(|\Sigma_Z|))}_{K(Both)} + \underbrace{\frac{1}{2}ln(|Var(Z|R)|)}_{V(I(R;Z))} + \underbrace{\frac{m}{2}(ln(2\pi) + 1)}_{D(H(R))}$$

## C    APPENDIX ANALYTICAL DETAILS

### C.1    LIMITATIONS AND BROADER IMPACT

Our work focuses on finding an optimal point between $H(R)$ and $I(R;Z)$ and analyzing the training dynamics that influence both terms. However, the notion of an "ideal" optimal point is difficult to prove with respect to a given data setting. Ideally, there should be a derived bound or ratio between $H(R)$ and $I(R;Z)$ that we can claim with high probability to correspond to an "ideal" optimal relationship between the two terms. Despite this limitation, the broader impact of this work is that it provides a general framework to develop SSL algorithms across diverse fields such as medicine (26), seismology (28), and autonomous driving (29). Therefore, it provides an avenue for potential growth of machine learning solutions in a wide variety of fields. We are unaware of any negative societal impacts directly caused by our work.

### C.2    VICREG VS. SIMCLR COMPARISON

The `AdaDim` methodology is based on the premise that VICReg better promotes higher $H(R)$ and the NCE loss used in SimCLR promotes lower $I(R;Z)$. These same dynamics are observed in a real SSL setting in Figure 12 where a ResNet-50 model is trained for 2000 epochs on Cifar-100 (31) using the VICReg and SimCLR SSL methods. In part a), both methods have an increase in $I(R;Z)$, but it occurs at a slower rate for SimCLR. In part b), the overall dimensionality of the dataset increases across all training epochs for $R$, but begins to plateau at the end of training, corresponding to the end of the feature decorrelation stage. $Z$ exhibits this same behavior, but plateaus much more noticeably throughout training which contributes to the plateauing effect of $I(R;Z)$. For both $R$ and $Z$, the overall dimensionality is lower for SimCLR than for VICReg. In part c), $R$ and $Z$ have a similar uniformity for both methods at the start of training, but significantly diverge from each other by the end of training.

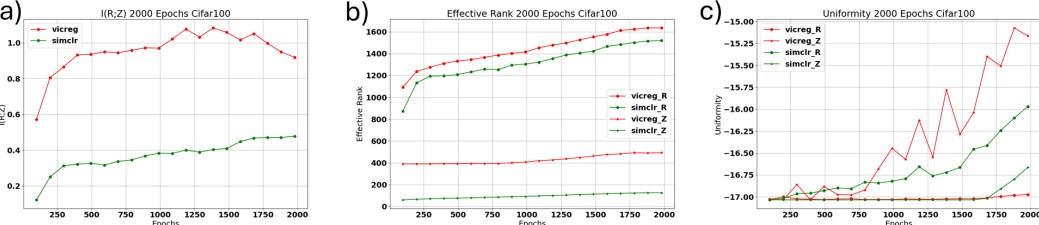

Figure 12: We train models with two different SSL methods for 2000 epochs and then analyze changes in a) $I(R;Z)$, b) effective rank between $R$ and $Z$, and c) uniformity between $R$ and $Z$.

These observed trends for SimCLR and VICReg hold for a wide variety of datasets in parts a) and b) of Figure 13. In part a), at the end of training for 6 different datasets, the $H(R)$ and $I(R;Z)$ values of VICReg are higher than those of SimCLR. In part b), these trends are analyzed over the course of manually setting the $\alpha$ parameter over the course of training from 0 to 1 in increments of 0.2. It is observed that as the optimization changes from VICReg ($\alpha = 0$) to SimCLR ($\alpha = 1$), the $I(R;Z)$ and $H(R)$ values monotonically decrease.

### C.3    GENERATION OF $H(R)$ VS $I(R;Z)$ PLOTS

In part c) of Figure 13, empirical results demonstrate the existence of an optimal point between high $H(R)$ and low $I(R;Z)$. The plots are generated by training each respective dataset with manually chosen $\alpha$ parameters within the `AdaDim` framework. In this case, manual means that the adaptive $\alpha$ computation is removed and a specific value of $\alpha$ from 0 to 1 is kept constant during training with the $\gamma$ parameter set to 0. These $\alpha$ values are $\alpha = [0, 0.2, 0.4, 0.5, 0.6, 0.8, 1.0]$. The logic behind this setup is to provide a controlled system where we can gradually transition between a feature decorrelation based loss and a sample uniformity based loss. The plot in part a) of Figure 1 was generated in a similar manner. However, in this case, the $\alpha$ increments are reduced to 0.05 In this way, we can more easily observe the emergence of the $H(R)$ and $I(R;Z)$ balance regions without needing to train as many models as is done in Figure 4.

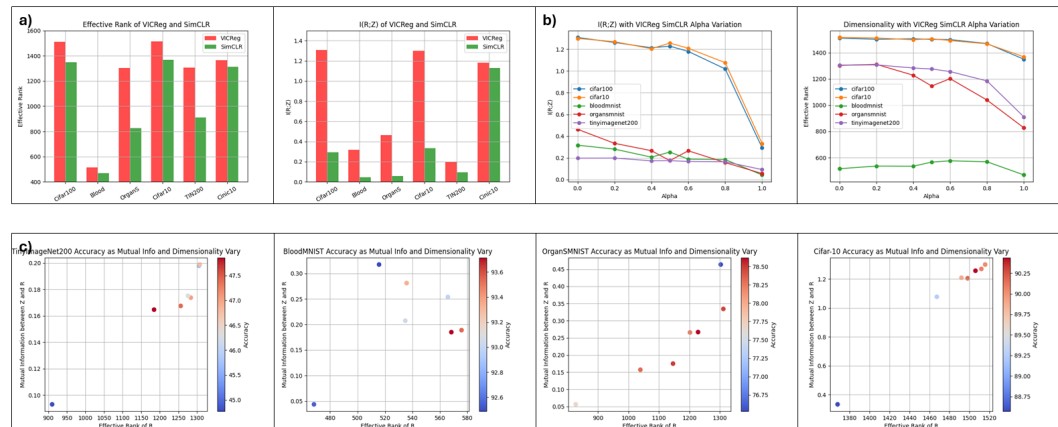

Figure 13: a) We compare the effective rank and $I(R; Z)$ between the representation of the test set for different datasets trained on VICReg and SimCLR. b) We show how the effective rank and $I(R; Z)$ vary after the introduction of the $\alpha$ parameter for each dataset. c) We show how performance varies as a function of $I(R; Z)$ and $H(R)$ for a variety of datasets.

More plots are provided for more datasets in part c) of Figure 13. Note that potentially more models need to be generated in order for the balance trend to be more salient for specific datasets. However, for many of these datasets, such as TinyImageNet, OrganSMNIST, and Cifar-10, the trend is clear that performance depends on a balance between $H(R)$ and $I(R; Z)$.

We also have experiments in which the $H(R)$ and $I(R; Z)$ plots are generated by completely random parameters. This is shown in Figure 4. In these cases, the trend of a balancing point is not as smooth, but it is clear that there exists a cluster of points around specific $H(R)$ and $I(R; Z)$ values that perform better than those points outside the discussed region. The exact parameters used to generate these graphs are provided in Table 13.

The same list of results is used to compute the correlation experiments in Table 1. In this table, we compute the results for a specified number of models and their corresponding results. We then compute the pearson correlation coefficient between the list of results for each model and the metric of interest that includes $H(R)$, $I(R; Z)$, or the ratio between them. In this case, the ratio is computed as simply $\frac{I(R;Z)}{H(R)}$. In the table, we report the magnitude of the correlation coefficient as we are only interested in the existence of a linear relationship between both terms, not whether it is positively or negatively correlated. In general, for all metric parameters, it is dataset dependent as to whether the correlation is positive or negative. This agrees with the analysis of our paper since each dataset has its own dimensionality characteristics and may favor a specific relationship between $H(R)$ and $I(R; Z)$.

### C.4 DISCUSSION OF RELATIONSHIP WITH LITERATURE

One surprising observation of this paper is that it opposes a variety of recent works (2; 45; 19). In these papers, the authors argue that some measure of dimensionality can be used as an unsupervised surrogate of representational quality. In other words, higher dimensionality should correspond to a better performing model on potentially any downstream task. However, our work suggests that both $H(R)$ and $I(R; Z)$ should be considered for an unsupervised assessment of model quality. However, our result is not surprising when we consider how these works justify their conclusions. For example, (18) based their rank estimates off of pre-trained ImageNet models which may not reflect the dynamics of training an SSL method on any given dataset. (45) showed a wide range of coefficient correlation values (0.2 - 0.8) between different dimensionality based metrics and performance values derived from various sources. This suggests that in some settings dimensionality is a good surrogate for performance while in others $I(R; Z)$ needs to be considered. This corresponds to the dynamics discussed in this paper, where the best performing model is often not the one with the highest dimensionality. It is the one that reaches a suitable intermediate point between both $H(R)$ and $I(R; Z)$. Our work suggests that future unsupervised estimators of representational quality should have some mechanism to detect this balance between the two terms of interest.

## C.5 MANUAL $\alpha$ USAGE

| Method | Alpha | Dataset | | | | | | |
| --- | --- | --- | --- | --- | --- | --- | --- | --- |
| | | Cifar100 | Cifar10 | TinyImageNet200 | Cinic10 | Blood | OrganS | iNat21 |
| SimCLR | N/A | 64.00 | 88.59 | 44.78 | 78.54 | 92.54 | 77.67 | 23.96 |
| VICReg | N/A | 64.70 | 90.02 | 45.54 | 78,25 | 92.48 | 76.50 | 24.24 |
| AdaDim | 0.2 | 65.18 | 90.07 | 46.75 | 78.27 | 93.36 | 78.41 | - |
| AdaDim | 0.4 | 66.15 | 90.18 | 47.00 | 78.57 | 93.04 | 78.46 | - |
| AdaDim | 0.5 | 66.53 | 90.43 | 46.26 | 79.35 | 92.98 | 78.50 | 24.56 |
| AdaDim | 0.6 | 66.11 | 89.87 | **48.06** | **79.58** | 93.56 | 78.23 | - |
| AdaDim | 0.8 | 66.32 | 89.25 | 47.83 | 78.54 | **93.71** | 78.26 | - |
| AdaDim | Ada | **66.90** | **90.72** | 47.81 | 79.53 | 92.86 | **78.55** | **24.81** |

Table 14: This shows the performance of AdaDim under different $\alpha$ parameters on several different datasets. $\gamma$ is set to 0 for this study.

In Table 14, an ablation study is performed where the $\alpha$ parameter is varied and the $\gamma$ parameter is set to 0. In this table, we use the symmetric augmentation scheme of (11) and pre-train each method for 400 epochs on a ResNet-50 model. This specific training setting is used as it allows us to better isolate changes in performance due to the specific optimization objectives and not due to other considerations such as the applied augmentation. We find that our adaptive methodology without $\gamma$ either outperforms or is consistent with the best result that we obtain from manually choosing a hyperparameter for $\alpha$. This highlights the importance of adaptively shifting between losses over the course of training to match the dynamics of SSL training.

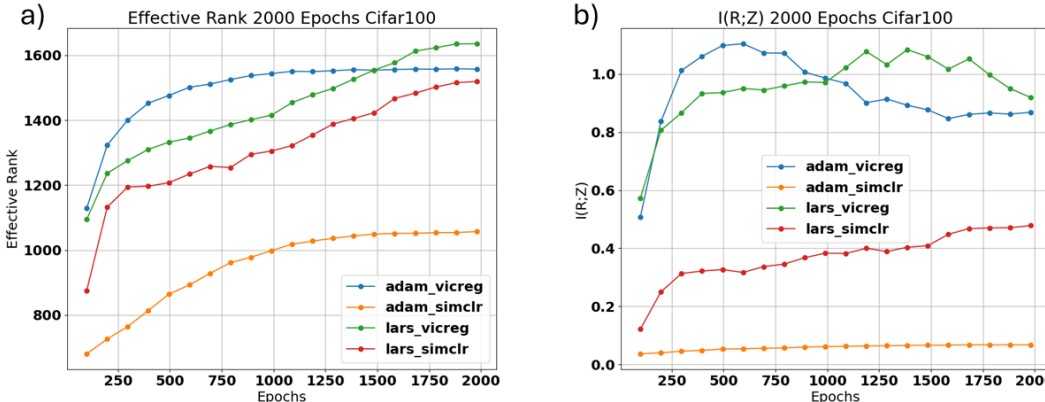

Figure 14: We show how a) $H(R)$ and b) $I(R; Z)$ varies for both SimCLR and VICReg under different optimization settings.

## C.6 VARIATION IN OPTIMIZATION PROCEDURE

In Figure 14, we vary the optimization setting for both SimCLR and VICReg. It is observed that the effective rank and $I(R; Z)$ curves have similar trends for both the adam and lars optimizers. However, the difference is that for the adam optimizer, the effective rank has a more pronounced upper limit on the values it can reach. Additionally, for $I(R; Z)$, the adam trained optimizer begins to decrease or plateau quicker. This result highlights that the trends of this paper are general, but its exact manifestation across training will vary based on the setup of the experiment.

## C.7 $\gamma$ ANALYSIS

In part a) of Figure 15, we perform an analysis of varying $\gamma$. We find that increasing $\gamma$ causes an increase in both $I(R; Z)$ and $H(R)$ while decreasing $\gamma$ causes the opposite effect. It is interesting to note that the same trend of an ideal balance emerges through the choice of the $\gamma$ parameter. In this case, performance is maximized when choosing $\gamma$ that results in more intermediate values between

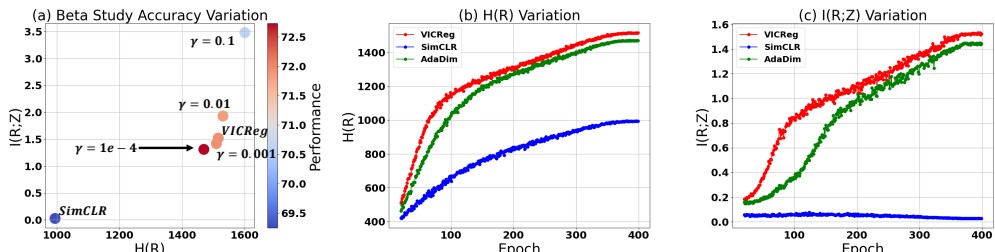

Figure 15: a) This figure shows how performance, $H(R)$, and $I(R;Z)$ change as $\gamma$ is varied. b) This figure shows how the effective rank of AdaDim varies compared to baseline methods. c) This figure shows how the $I(R;Z)$ for AdaDim varies compared to baseline methods.

| ViT Cifar-100 Comparison | | |
|---|---|---|
| Method | Swin-Small (33) | ViT-Large (14) |
| SimCLR | 53.91 | 63.92 |
| VICReg | 56.83 | 63.91 |
| AdaDim | **58.07** | **66.35** |

Table 15: This table shows the the performance of AdaDim under a vision transformer setting (33; 14).

both $H(R)$ and $I(R;Z)$. This illustrates the potential importance of tuning $\gamma$ for specific data settings as each dataset will have a specific balance between $H(R)$ and $I(R;Z)$ that is optimal.

In parts b) and c), we compare the growth of $H(R)$ and $I(R;Z)$ over the course of training a ResNet-50 model for 400 epochs on Cifar-100. We find that the adaptive regularization technique of our method leads to `AdaDim` reaching a trajectory that attempts to balance between both terms when compared with the more biased trajectories of SimCLR and VICReg.

We also note that the trajectory of AdaDim follows a similar trajectory to that of VICReg. In the design of our method, VICReg is used much more heavily at the start of training which is when much of the structure of the representation space is learnt. However, the gradual introduction of the NCE loss of SimCLR is an important aspect gradually introduce sample uniformity at the end of training. This importance is reflected in the significant performance improvements throughout the paper even if its introduction only lowers the corresponding H(R) and I(R;Z) values slightly.

## C.8 VISION TRANSFORMER EXPERIMENTS

In Table 15, we provide comparisons of AdaDim when trained with a vision transformer backbone. Specifically, we perform a comprehensive study of a smaller scale transfomer (33) and a larger scale ViT model (14). We train these networks on Cifar-100 for 400 epochs, with an AdamW optimizer, batch size of 256, learning rate of 2e-4, weight decay of 0.05, and betas of 0.9 to 0.95. For these experiments, we use the asymmetric augmentation setting. We find that AdaDim out performs standard VICReg and SimCLR baselines for both models. The reason for this improvement may be due to specific properties regarding transformers. Specifically, transformers benefit from longer training times compared to traditional CNNs. When training with more epochs, there is a greater ability for the model to go through the training dynamics dynamics discussed in this paper. It is possible that AdaDim is advantageous in this setting due to providing explicit regularization based on these emergent dynamics.

## C.9 COMPARISON AGAINST ALTERNATE TRAINING STRATEGIES

In Table 2, we provide an ablation study of the impact of the different components of our method and hypothetical ways our method could be applied. This includes strategies such as a fixed $\beta$ parameter during training and a heuristic scheduler of the $\alpha$ parameter. Possible schedules include cosine or linear scaling from 0 to 1. To further validate the importance of adaptive scaling of both

| AdaDim Standardized Hyperparameter Ablation Study | | | | | | | | |
|---|---|---|---|---|---|---|---|---|
| Method | $\alpha$ Type | Cifar100 | Cifar10 | TinyImageNet200 | Cinic10 | Blood | OrganS | iNat21 |
| SimCLR (7) | N/A | 64.00 | 88.59 | 44.78 | 78.54 | 92.54 | 77.67 | 23.96 |
| VICReg (3) | N/A | 64.70 | 90.02 | 45.54 | 78.25 | 92.48 | 76.50 | 24.24 |
| SimCLR + $\lambda$ (35) | N/A | 64.37 | 88.00 | 45.54 | 76.96 | 92.86 | 77.98 | 23.51 |
| VICReg + $\lambda$ (35) | N/A | 64.54 | 89.77 | 45.83 | 78.47 | 92.43 | 77.16 | 24.01 |
| SimCLR +VICReg | $\alpha = 0.5$ | 66.53 | 90.43 | 46.26 | 79.35 | 92.86 | 78.50 | 24..56 |
| SimCLR +VICReg | cosine | 65.78 | 88.85 | 45.45 | 78.87 | 92.57 | 78.46 | - |
| SimCLR +VICReg | linear | 66.99 | 89.53 | 45.94 | 78.60 | 92.07 | 78.61 | - |
| AdaDim ($\alpha$ = Ada) | Ours | 66.90 | 90.72 | 47.81 | 78.55 | 93.10 | 78.55 | - |
| AdaDim ($\alpha$ = Ada, $\beta$ = Ada) | Ours | **67.15** | **90.81** | **48.24** | **79.53** | **93.24** | **79.19** | **24.81** |

Table 16: This table shows an ablation study of performance across a variety of datasets when varying aspects of the construction of the AdaDim loss.

| AdaDim Batch Size Comparison Experiments | | |
|---|---|---|
| Batch Size | ImageNet-100 + LARS | Cifar-100 + SGD + Momentum |
| 128 | 79.30 | - |
| 256 | 79.64 | 74.31 |
| 512 | 80.06 | 71.67 |
| 1024 | - | 71.02 |

Table 17: In this table, we compare the performance as batch sized is varied for two different experimental settings. Both methods are trained with a ResNet-18 encoder. ImageNet-100 experiments are trained for 400 epochs while Cifar-100 experiments are trained for 1000 epochs. For these experiments, the $\gamma$ parameter is kept fixed at 1e-4. ImageNet-100 uses the baseline AdaDim strategy, while Cifar-100 is trained with the AdaDim + momentum method.

the $\alpha$ and $\beta$ terms, we provide a comprehensive study of these terms under a fixed hyperparameter setting in Table 16. The experiments in this table are setup such that each method has the exact same hyperparameters to ensure that any deviations in performance are caused solely by the optimization objective. This includes 400 epochs of training with a ResNet-50 model, the symmetric augmentation scheme of (11), a LARS optimizer, learning rate of 0.4, and a batch size of 256 for all experiments. Again, we observe that our method that makes use of a dimensionality adaptation technique out performs heuristic schedules as well as methods based on fixed regularization of $I(R; Z)$ (35) across a wide variety of data settings.

## C.10 BATCH SIZE DISCUSSION

In the main paper, we conduct all our experiments with small batch sizes of 128 or 256 depending on the context of the experiment. In all cases, we achieve state of the art performance without necessitating large batch sizes as discussed in (7). However, we also wanted to assess how our method would perform in large batch settings. We perform this analysis in Table 17 where we compare the performance of our method in two divergent settings. This includes the baseline hyperparameter setting with ImageNet-100 and the expanded hyperparameter setting with a momentum encoder trained on Cifar-100. In the case of ImageNet-100 with the LARS optimizer, we find that performance improves with larger batch sizes. However, with Cifar-100 and the SGD optimizer, performance degrades with the introduction of larger batch sizes. We believe the reaction to batch size is largely a matter of identifying optimal optimization settings for the large batch setting. For example, ImageNet-100 possibly improved due to the LARS optimizer being better suited to scaling optimization with respect to the large batch setting. However, for the SGD setting, further tuning of learning rates and parameters such as $\gamma$ may be needed for effective downstream performance on large batches. Additionally, the batch setting improvements may also be dataset dependent with ImageNet-100 benefiting much more from large batches than Cifar-100. This may be the case as most batch size results in SSL papers (7; 8) are benchmarked on large scale datasets such as ImageNet, rather than Cifar-100. Regardless, we see the small batch size state of the art performance of our method as a major advantage compared to the resource intensive nature of other methods.

