# OpenReview forum: "AdaDim: Dimensionality Adaptation for SSL Representational Dynamics"
_ICLR.cc/2026/Conference — Submitted to ICLR 2026_

### Official Review · Reviewer_ZhAj · 2025-10-27

**Soundness:** 3
**Presentation:** 1
**Contribution:** 2
**Rating:** 2
**Confidence:** 3

**Summary:**

This paper explores the interaction between some components in a training loss for self-supervised learning (SSL), including the entropy of a representation and mutual information between the representation and projection embedding. The authors analyze the dynamics of those quantities along the training process, and observed some insights from both simulation and real datasets. From those insights, the authors propose to add two constants to the training loss, which can balance different properties of the trained representation. Finally, the authors evaluate their proposal and different SSL methods on standard benchmark datasets. The results suggest that their proposal can be competitive or better than those SSL methods.

**Strengths:**

- The authors demonstrate the clear interaction between the entropy and mutual information, which indicates some properties of the learned representations. Those interactions are reported to appear for popular SSL methods.
- Their proposed balancing method was compared with different recent SSL methods. The method seems to be comparable or better than the baselines.

**Weaknesses:**

- The paper lacks lots of details: including important notations and experimentation setup. For instance,
    - The key notations are not defined explicitly, e.g.,
        - The entropy $H(R)$, mutual information $I(R,Z)$, the determinant, ...
        - The key functions in their proposed loss: $s, v, c$
        - Constants $\lambda, \mu, \nu$, ...
    - PCA was used to visualize the dynamics. However, it is unclear about how the authors can ensure the accuracy of the dynamics when projecting onto a lower dimensional space by PCA. A control on this accuracy is important to investigate a dynamic in high-dimensional spaces.
    - Some figures (and investigation) reports the eigenvalues of *unclear* matrices/objects.
- The proposed loss introduces two more parameters, leading to too many parameters for their method and hence complicating model selection.
- The experiments use ResNet18 and ResNet50, which are classic architectures. More recent (stronger) architectures should be investigated.
- The overall writing is not very clear. Some figures (fig. 4) and tables (tab. 1, 3, 4, 5) are too small. Some typos remain, even for the paper title. Inconsistent notations remain, e.g., $ln$ and $log$.

**Questions:**

N/A

---

> ### Author Response · Authors · 2025-11-15
> **Initial Response to Weaknesses and Questions Part 1**
>
> Thank you reviewer ZhAj for taking the time to evaluate our work! We look forward to in-depth discussions over the next few weeks.
>
> We are happy to clarify the confusion over different notations from the main paper, but we would like to point out where these different conventions are located in the main paper. To organize this discussion, I will respond to each of your points in the weakness section in the order in which they appear in your open review comment.
>
>
>
>
>
> **Weakness 1:**
>
> To address this clarity issue, I will clarify where the key components of our notations can be found in the main paper as well as the original work that the notation is cited from.
>
> In  line 46, we identify that we measure H(R) through the effective rank metric where the details of this metric can be found at paper [35] in our citations. Additionally, in section A.3 of the appendix, we have a discussion about the metrics used in the analysis for both H(R) and I(R;Z) and where the choice of these metrics came from and the exact intuition for each. With respect to I(R;Z), we describe in this section that we are using the alpha-Renyi matrix mutual information estimator to calculate this value and the details of this estimator can be found in citation [40] and is written out in A.3 explicitly. We also describe how we calculate I(R;Z) in line 350 of the methodology section. However, we agree that we can make these points more salient as to how to identify the calculation.
>
> With respect to s, mu, and c in the proposed loss, these are the components of the vicreg loss that we are using in our method as a surrogate for a decorrelation based function. VICReg is a standard SSL loss that we cite in the main text [3]. We did not have enough space to go into the details of the VICReg losses parameters in the main text, but did state on line 318 that the full details of this loss can be found in section A.5 of the appendix. This is also true of lambda, mu, and v. These are all constants associated with the VICReg loss. Full details of all these terms can also be found in the original vicreg paper which is citation [3] in the paper. Since we have more space for the revision, we can potentially include this directly in the main paper.
>
> **Weakness 2:**
>
> I also saw an issue with us using PCA to simulate the dynamics. A common technique in the analysis of Deep Learning models is to make a gaussian assumption on the data in order to study the theoretical considerations on the data space. This is also common within the context of SSL research with works such as:
>
> Shwartz-Ziv, R., Balestriero, R., Kawaguchi, K., Rudner, T. G., & LeCun, Y. An information-theoretic perspective on varianceinvariance-covariance regularization. arXiv 2023. arXiv preprint arXiv:2303.00633.
>
> I also make a gaussian assumption on the data space during the simulation of Section 3. This allows me to isolate the analysis in terms of terms that can be readily understood such as the variance of the data space. In this setting, I want to study the relationship between the original data space and its projection. Since the data space I am studying for this theoretical study is gaussian, this means much of the information associated with the data is encoded by its variance parameter. Therefore, in this setting PCA serves as a valid projection because the whole idea behind PCA is to preserve as much of the variance of the original data space as possible. This analysis allows me to get a sense of the dynamics that exist under a gaussian assumption on the data. This reasoning is described partially on line 193, but I can include more details in the revision of the paper.
>
> However, I agree with the reviewer that the Gaussian assumption may not hold in higher dimensional spaces such as a deep neural network. It is for this reason, we show that the same trends hold when the projector is a neural network as mentioned on line 195 where a simulation exists in section B.4 and B.5 of the appendix. We also empirically study these same trends within the context fn a standard SSL deep learning setting in Section 3.2. In these figures, we show how the feature decorrelation and sample uniformity manifest in real training settings and that it follows similar trends to those that we discuss with a gaussian simulation in part 3.1.
>
> **Weakness 3**
>
> We also note the exact matrices that our eigenvalue analysis was performed on in Section A.3 of the appendix where we go into full analytical detail of these analytical experiments. In this section, we describe how the eigenvalues in these analyses of Section 3.2 are performed on the matrix formed from output representations of the test set for each dataset of interest in the study. This matrix has a size of test set size * feature dimension size. These details can be more explicit in the main paper for the revision in order to clarify any confusion.

---

> ### Author Response · Authors · 2025-11-15
> **Initial Response to Weaknesses and Questions Part 2**
>
> **Weakness 4:**
>
> Another discussed weakness of our method is the introduction of “two new parameters.” We introduce a single new parameter which is the gamma parameter. This is because the alpha and beta parameters are computed adaptively based on the dimensionality characteristics of the data of interest. It should be noted that a single additional parameter, while not ideal, is not too different from the variety of parameters that are regularly introduced with different SSL methods. A central reason for this is that SSL doesn’t have a defined labeled target and specific tuning of parameters regularly shows up in state of the art approaches due to this uncertainty about the target data space of interest such as the weighting on variance, covariance, and invariance on the VICReg method and the temperature parameter in SimCLR. We would also like to note that in Table 2, we perform a comprehensive study of these different parameters to hypothetically test the performance of alpha and beta compared to being adaptively calculated as is done in the base method.
>
> Additionally, we try to maintain the gamma parameter at 1e-4 for almost every experiment in the paper besides the ImageNet-100 comparison to demonstrate only minimal tuning is necessary for comparable state of the art performance. However, we show that tuning gamma on a dataset specific basis causes further performance gains above comparable state of the art methods in the new version of Table 3. This highlights the adaptability of our method to conform to the $I(R;Z)$ and $H(R)$ trajectories for potentially any dataset.
>
> **Weakness 5:**
>
> The reason that we use ResNet-18 and ResNet-50 is because most baselines in SSL use these architectures. Therefore, specific hyperparameter and optimization choices for every SSL method exist for these architectures, but not necessarily for new methods such as vision transformers. However, we do agree that for experimental robustness, we should include some type of vision transformer comparison. We will be sure to add this into the revision that we will upload soon.
>
> **Weakness 6:**
>
> We were under space constraints in the original version of the paper and are happy to increase the size of every table and figure given that we are able to use  an extra page of space for the revision. We hope that the above discussions can provide clarity for you on the major impact of the paper which is characterizing previously not understood training dynamics of SSL approaches and then devising a method to take advantage of these identified dynamics. We will also be sure to be on the look out for typos that we can improve in the rebuttal revision of the paper.

---

> > ### Comment · Reviewer_ZhAj · 2025-11-27
> >
> > I thank the authors for providing some explanations. However, the submission has been the same until now. In my opinion, all notations should be introduced clearly before their use, and the authors should not ask the readers to read some other papers to know the notations.
> >
> > I still keep the score unchanged.

---

> > > ### Author Response · Authors · 2025-11-27
> > >
> > > Thanks for the response, we will be releasing the updated version of the paper by the end of the week.

---

> ### Author Response · Authors · 2025-11-28
> **Revised Paper Update**
>
> **Weakness 1 Revised Paper Update:**
>
> We have taken additional efforts to improve the clarity of the entire paper. Specifically, we now provide detailed pseudocode of our method in Section A.3 and further experimental details in Section A.4. We have also included more citations and links in the text to where specific details can be found. We also highlighted Section A.5 of the paper where a full discussion of the VICReg and NCE losses can be found.
>
> **Weakness 2 Revised Paper Update:**
>
> We provide more details in Section 3 to justify the usage of the PCA projection and cite other papers where a gaussian analysis is performed in a PCA setting. Furthermore, we do a better job of also highlighting the location of our Neural network equivalent simulation in Section B.4.
>
>
> **Weakness 3 Revised Paper Update:**
>
> We now include an explicit explanation of the matrices used in the calculation of the analysis of Figure 3 in the main text. We also provide a reference to the section where all calculations are explained in detail in both the introduction with Section A.6 and under Figure 3 with Section A.9.
>
> **Weakness 4 Revised Paper Update:**
>
> We added additional clarifying sentences in the main paper to specify that our method uses a single hyperparameter. Additionally, we clarify that the analysis of Table 2 provides a hypothetical ablation study of scenarios where the parameters are potentially chosen as fixed values.
>
> **Weakness 5 Revised Paper Update:**
>
> We provide vision transformer ablation studies in Section C.8.
>
> **Weakness 6 Revised Paper Update:**
>
> We have taken efforts across the paper to improve the quality of writing in dense sections as well as space out figures and tables to a better degree for visual quality. We hope these changes make it easier for the contributions of our work to come out.
>
>
> We also want to mention that we have performed additional experiments to improve the experimental robustness of the entire paper in addition to addressing your weaknesses above. This is reflected in the new Tables 3, 4, and 5. Specifically, we now have two settings: AdaDim with baseline parameters and AdaDim with parameters suitable for leveraging an additional momentum encoder model. We find that the baseline AdaDim setting can improve performance with dataset specific tuning of the gamma parameter and is comparable to methods to make use of a variety of architectural tricks. However, we also found that AdaDim leverages a momentum encoder particularly well to the point that it exhibits state of the art performance in both the solo learn and ImageNet 100 epoch benchmarks. We also analyzed the experimental robustness across different amounts of training time  in Figure 6.
>
> Finally, we provide a compute cost analysis of our method in section A.8 to demonstrate that our method does not introduce prohibitively expensive computational costs into the optimization setup.

---

### Official Review · Reviewer_MV6A · 2025-10-29

**Soundness:** 3
**Presentation:** 3
**Contribution:** 3
**Rating:** 4
**Confidence:** 4

**Summary:**

This paper studies how the dimensionality of SSL representations H(R) and the mutual information between representation and embedding spaces  I(R;Z) affect the learning performance. The authors argue that many previous works believe higher H(R) and lower I(R;Z) always lead to better results, but this is not always true.
They show, both theoretically and experimentally, that the best performance happens when there is a balance between these two terms. Based on this finding, they propose AdaDim, a training strategy that adaptively adjusts between feature decorrelation and sample uniformity while gradually regularizing I(R;Z). The method improves SSL performance by about 1–3% on several datasets without using extra modules such as momentum encoders or teacher–student networks.

**Strengths:**

- The paper brings a new view to self-supervised learning by linking representation dimensionality and mutual information. It gives both mathematical and empirical evidence to support this idea.

- AdaDim is lightweight and easy to apply. It only changes the loss weighting dynamically and does not require any architectural modification.

- The authors test on CIFAR, Tiny-ImageNet, medical datasets, and ImageNet-100. The method shows consistent improvement across settings.

- The writing is well organized, and the results in tables and figures are easy to follow.

**Weaknesses:**

The reviewer's main concern is about the experiments:
- The reported improvement in Table 4&5 is marginal.
- The paper does not study how performance changes with different batch sizes or temperature values. These are well-known factors that strongly affect contrastive learning. Larger batch sizes might reduce the improvement effect.
- The method is tested only on small or medium datasets. Results on larger benchmarks such as full ImageNet would better support the generalization claim.

Minor formatting issue - There is no space above section 6 title and it seems to be squeezed in for page requirement.

**Questions:**

My main concern should refer to the weakness section, and I am willing to change the rating if the weakness is being explained or addressed.

---

> ### Author Response · Authors · 2025-11-15
> **Initial Response to Weaknesses and Questions Part 1**
>
> Thank you reviewer MV6A for taking the time to evaluate our work! We look forward to in-depth discussions over the next few weeks.
>
> We agree that additional experiments to probe the robustness of our method would help reinforce our claims. This will be added when we generate the revision of our paper very soon. However, we would like to start discussion by commenting on your currently existing points. I will number the weaknesses based on the order in which they are discussed in your comment.
>
> **Weakness 1:**
>
> In Tables 4 and 5, we report performance against the other methods tuned for better performance based on the solo-learn library of the original papers. In the vast majority of cases, our method is still able to out perform competing baselines without the need for any additional architectural tricks. Our gains are strictly from carefully adapting losses based on the training dynamics that we introduce as a major contribution. In some cases, this results in significant performance gains as in Cifar-100, ImageNet-100, TinyImageNet200, bloodmnist, and organsmnist, but in other cases the gains are not as significant. One important consideration is that for all the comparisons in Table 5, we operate under a fixed gamma value of 1e-4 for consistency of comparison. However, it may be the case that additional tuning of this gamma parameter might be more beneficial to other datasets. Again this is something that we can address with additional experimentation during the revision rebuttal period.
>
> **Weakness 2:**
>
> In Figure 16 in Section C of the appendix, we perform various ablation studies on projector output size, temperature, and the effect of changing the update on our alpha parameter. In all cases, we find no significant performance degradation. We are also happy to include additional studies on batch size in the revision of the paper.
>
> **Weakness 3:**
>
> We agree that we only have small and medium sized datasets. We are currently running an ImageNet evaluation for the revision of the paper. However, we will need some time due to limited computation to get a sense of the evaluation.
>
> We can also fix the minor formatting issues you described.

---

> > ### Comment · Reviewer_MV6A · 2025-11-26
> >
> > Thank you for the explanation. Based on the current experimental results, I will maintain my current rating, as weaknesses 1 and 3 have not yet been addressed. I am open to reconsidering the rating once updated results are available.

---

> > > ### Author Response · Authors · 2025-11-27
> > >
> > > Thanks for the response, we will be releasing the updated version of the paper by the end of the week.

---

> ### Author Response · Authors · 2025-11-28
> **Revised Paper Update**
>
> **Weakness 1 Revised Paper Update:**
>
> We have performed additional experiments to improve the experimental robustness of the entire paper. This is reflected in the new Tables 3, 4, and 5. Specifically, we now have two settings: AdaDim with baseline parameters and AdaDim with parameters suitable for leveraging an additional momentum encoder model. We find that the baseline AdaDim setting can improve performance with dataset specific tuning of the gamma parameter and is comparable to methods to make use of a variety of architectural tricks. However, we also found that AdaDim leverages a momentum encoder particularly well to the point that it exhibits state of the art performance in both the solo learn and ImageNet 100 epoch benchmarks. We also analyzed the experimental robustness across different amounts of training time  in Figure 6.
>
>
> **Weakness 2 Revised Paper Update:**
>
> We performed a new comprehensive ablation study across many different hyperparameters in Figure 7 of the main paper. We also provide a batch size analysis in section C.10.
>
> **Weakness 3 Revised Paper Update:**
>
> We now have a comprehensive benchmark evaluation of our method on ImageNet in Table 5 of the main paper.

---

### Official Review · Reviewer_ACRX · 2025-11-01

**Soundness:** 2
**Presentation:** 1
**Contribution:** 1
**Rating:** 2
**Confidence:** 4

**Summary:**

The authors propose to study the training dynamics of SSL models by measuring the dimension of the representations (H(R)) as well as how much the projector affects I(R;Z). Showing that commonly used SSL methods have different training dynamics for those metrics, the authors posit that a better model can be obtained by finding the sweet spot between H(R) and I(R;Z), combining existing methods to do so. An additional loss term to help maximize I(R;Z) is introduced and a partially automated approach is introduced to simplify the hyperparameter search. Extensive experiments are conducted and an increase in performance is observed over the baselines.

**Strengths:**

- Incorporating I(R;Z) ( which is related to the degree of invariance of the representations) with the anti-collapse is useful. Similar ideas were studied in Figure 4 of [1] which could help enrich the discussion.

- The analysis focusing on training dynamics is appreciated, as previous works mainly focus on the loss itself or the final representations.

- The analysis of I(Y;R) is interesting, although since it is an upper and not lower bound it should be interpreted cautiously.

- The introduction of multiple hyperparameters is mitigated by a partially automated strategy which simplifies the hyperparameter search.

- In controlled settings, the authors demonstrate performance gains over baselines.

[1] Li, Alexander C., Alexei A. Efros, and Deepak Pathak. "Understanding collapse in non-contrastive siamese representation learning." ECCV 2022.

**Weaknesses:**

1) While performance gains are visible with standardized hyperparameters, in more ideal setting (where every method is tuned to the best of their ability) gains are more marginal

2) Misrepresentation of previous work line 85. [41](from the paper's numbering) Does use the degree of invariance of models to augmentations, which can be related to I(R;Z). [17](from the paper's numbering) also does not measure H(R) but H(Z), which can be related to both H(R) and I(R;Z).

3) From Figure 6, it seems that AdaDim has dynamics for H(R) and I(R;Z) that are very close to VICReg, which puts into question the real gain of using SimCLR.

4) The method appears overly complex for marginal gains over VICReg. While VICReg and SimCLR have different approaches to avoid collapse, they can be isolated from the loss (Variance and Covariance loss for VICReg, logsumexp of negative pair distances for SimCLR). Using those directly would lead to a simpler loss, removing the presence of the invariance loss term in each sub-loss.

5) The goal of $L_{mut}$ seems unclear. If the goal is to maximize I(R;Z), couldn't this be done with improved weighting ? Looking at Figure 6, VICReg already has this I(R;Z) maximization effect. If it is mainly moderating the effect of SimCLR on I(R;Z), it again feels more like a weighing issue.

6) From prior work (e.g [2]), VICReg and SimCLR losses have very similar goals but with different dynamics, as illustrated in the current work.Notably, VICReg's loss is a much stronger regularization, and SimCLR a weaker one (due to the logsumexp). Combining the two losses is a way to get the benefits of both as highlighted in this work.
However, saying that the method proposed here interpolates between sample and dimension contrastive methods (lines 133-135) directly contradicts [2], especially theorem 3.3. This should be clarified in the paper.

[2] Garrido, Quentin, et al. "On the duality between contrastive and non-contrastive self-supervised learning." ICLR 2023.

**Questions:**

1) In table 3, why is the performance on CIFAR100 much lower for all methods than in table 2 ? From the text (line 416) it seems that both tables are done in the same setup, which should lead to comparable numbers.

2) $\beta$ is defined as $\gamma \times \alpha$ line 355, so how is it possible to have $\alpha$, $\beta$ and $\gamma$ all altered independently in Table 2 ?

4) To update $\alpha$, ER(Z) is estimated by using a batch of size 256 (line 345-346) but the dimensionality of Z is 2048. This means that ER(Z) is bounded by the batch size which makes it a poor estimator the effective dimension of Z. Did you experiment with other strategies using larger batch sizes for the ER(Z) computation ?

---

> ### Author Response · Authors · 2025-11-15
> **Initial Response to Weaknesses and Questions Part 1**
>
> Thank you reviewer ACRX for taking the time to evaluate our work! We look forward to in-depth discussions over the next few weeks.
>
> We would also like to point out that we are currently working on the revision that will be uploaded soon and will further enrich our discussion.
>
> **Weakness 1:**
>
> It is true that the gap in performance improvement decreases between tables 3 and 4. One of the major challenges with SSL research is that different methods respond differently to different hyperparameter  and augmentation choices. In an effort to create a fair comparison between all relevant literature, we decided to make two tables: the first where all methods receive the exact same set of hyperparameters and augmentations and a second table where all methods received the default tuned hyperparameters and augmentations of the solo-learn codebase across a variety of datasets. In this setting, where every method receives their best tuned parameters, it is expected that each method should perform better. For this reason, our method being competitive with each method tuned to better parameters highlights the generality of our method and we view it as a positive result.
> However, it is true that additional experiments and tuning would help reinforce our paper from an experimental point of view. For this reason, we will include additional results in a variety of settings in the revised version of the paper that we will upload soon.
>
> **Weakness 2:**
>
> This weakness relates to improperly describing previous work on Line 85. Our purpose with this line is to describe methods that are designing some metric of dimensionality in order to assess the quality of a representation space. [41] and [17] both devise a measure of dimensionality and argue that, even without fine-tuning, the score they arrive at better correlates with downstream accuracy. In fact, both of their methods end up using the effective rank metric that we use to represent H(R) at some point during the calculation of their method. [41] performs the effective rank calculation on the matrix resulting from a linear discriminant analysis calculation as seen in equation 4 of their paper. [17] performs the effective rank calculation on the matrix at the output of the projector H(Z) as you described. We did not intend to claim that these papers measured H(R) directly during the computation of their method. However, their methods do implicitly lead to a measurement of H(R) through the effective rank calculation on matrices that are either projections or alterations of the original representation space R. Thus, we characterized these works as suggesting that higher entropy is the major concern for downstream transferability of SSL models.
>
> We would also like to point out that we have a comprehensive discussion of the difference between our work and these works in Section C.4 of the appendix. Specifically, our work opposes [17] and [41] by stating that measurements of $H(R)$ are not enough and true downstream transferability needs to consider both $H(R)$ and $I(R;Z)$. In this section, we explain exactly why our conclusion differs and it has mainly to do with the experimental setting of these other works. [41] indirectly motivates our work by showing a range of correlation values from 0.2 to 0.8 that indicates $H(R)$ has some relationship with performance, but something else is needed (i.e. $I(R;Z)) to make this correlation stronger. [41] also revealed that the correlations of [17] were not as strong as appeared in their paper possibly because all their model relied on pre-trained ImageNet encoders rather than an analysis of the overall dynamics that appear over the course of training as is done in our work.

---

> > ### Author Response · Authors · 2025-11-15
> > **Initial Response to Weaknesses and Questions Part 2**
> >
> > **Weakness 3:**
> >
> > We believe one of the most significant contributions of our work is characterizing the training dynamics of SSL approaches. We then devise a method that is able to reflect the underlying dynamics that we observed in Section 3. One major point of our method is weighting the decorrelation loss (VICReg) much more heavily at the start of training compared to the sample uniformity loss (SimCLR). A lot of the basic representational structure of a model is learnt at the start of training and the end of training is reserved for fine-tuning. This is oftentimes referred to as a fitting phase followed by a compression phase in other work [1]. However, how these dynamics exactly manifest empirically in SSL wasn’t previously explored before our paper. We argue for the existence of a decorrelation phase at the beginning where the model learns as many features as possible by projecting into a higher dimensional space followed by a  sample uniformity phase where the model reorganizes its samples and lowers I(R;Z).
> >
> > In the case of training with our method, VICReg is used much more heavily at the start of training which could possibly cause the model to follow a similar trend as in Figure 6. However, the gradual introduction with the NCE loss of SimCLR is an important aspect as well to reflect the sample uniformity at the end of training even if lowering the corresponding H(R) and I(R;Z) values only slightly by its introduction brings the model closer to a point that better balances both. Additionally, the gradual introduction of SimCLR reflects the observed training dynamics of Section 3 where uniformity at the end of training contributes to a lower I(R;Z).
> >
> > This is empirically demonstrated in our Table 1 where VICReg (72.14) alone (alpha =0, beta = 0, gamma = 0) performs worse than SimCLR + VICReg (72.30) together (alpha = Ada, beta = 0, gamma=0) which performs much worse  than all the terms together (72.73). This is further verified in table 2 where we perform a similar ablation study of SimCLR alone, VICReg alone, SimCLR + VICReg weighted equally, SimCLR + VICReg weighted adaptively, and the final version of our method that incorporates L_mut.
> >
> > [1] Shwartz-Ziv, R., & Tishby, N. (2017). Opening the black box of deep neural networks via information. arXiv preprint arXiv:1703.00810.
> >
> > **Weakness 4:**
> >
> > It is true that potentially we could have used just the variance and covariance loss alone with VICReg and the sample uniformity alone with the SimCLR loss and then removed the invariance term. In fact, we tried this, but we found that the representations would quickly collapse. The reason for this is because the invariance loss of SimCLR and VICReg are not operating on the same embedding space. SimCLR operates on normalized embeddings while VICReg is trained on unnormalized inputs. VICReg relies on the variance loss term to do an implicit normalization, but this difference requires that the invariance term for each type of collapse mitigation (decorrelation vs. sample uniformity) to be maintained for the overall loss function.
> >
> > For the revised version during this rebuttal period, we are happy to make this distinction more salient.
> >
> > **Weakness 5:**
> >
> > The goal of  L_mut is to complete the incorporation of the dynamics that we described in Section 3. Specifically, we show in Figure 3 that for a wide variety of different methods, I(R;Z) increases at the start of training and decreases at the end of training. This means that in order to obtain balance between H(R) and I(R;Z) by the end of training, we want to gradually introduce additional I(R;Z) regularization to counter the decrease in I(R;Z) that is a natural aspect of SSL training dynamics. This reasoning is described in part d) of Figure 5 where the natural trajectory of the SSL dynamics follows increasing I(R;Z) followed by decreasing I(R;Z). Therefore the regularization  term on L_mut gradually increases over the course of training in order to counter the decrease in I(R;Z) that results at the end of training.
> >
> > Empirically, in Table 2 we also show the difference between keeping the L_mut term fixed is a substantial difference in performance compared to when it grows adaptively over the course of training.
> > We can make this point more salient in the revised version of the rebuttal.

---

> > > ### Author Response · Authors · 2025-11-15
> > > **Initial Response to Weaknesses and Questions Part 3**
> > >
> > > **Weakness 6:**
> > >
> > > We agree with the reviewer that we should clarify our writing in order to better contextualize our contribution with respect to [18] in the main paper. To start, when we said “interpolation” we meant between the dynamics expressed by a feature decorrelation approach such as the dimension contrastive methods and the sample uniformity methods expressed by sample contrastive strategies. We are happy to update the writing to avoid this confusion. As the reviewer discussed, our intuition is to get the benefits suggested by each approach. However, we argue that those “benefits” relate to the specific training dynamics about how each approach influences the resulting H(R) and I(R;Z) during training. Theorem 3.3 in [18] describes conditions under which sample and dimension contrastive approaches are equivalent. However, the authors of the work also state:
> > >
> > > “While theorem 3.3 shows that sample-contrastive and dimension-contrastive approaches minimize similar criteria, for none of these methods can we conclude that both criteria can be used interchangeably.” It then goes into a discussion of how a major difference between the approaches is the manner in which the embeddings are normalized as we discussed in Weakness 4.
> > >
> > > In fact, more recent work [2] (below), identified the possibility of developing methods that integrate the differing entropy estimation properties of VICReg and SimCLR to gain the advantage of both. (Section 5.1 of their paper). We believe our paper provides a more in depth distinction of the training dynamics that separate these two methods and the intuition behind which to properly integrate both during training.
> > > We are happy to provide a clearer discussion of these points in a revised version of the submission.
> > > [2] Shwartz-Ziv, R., Balestriero, R., Kawaguchi, K., Rudner, T. G., & LeCun, Y. An information-theoretic perspective on varianceinvariance-covariance regularization. arXiv 2023. arXiv preprint arXiv:2303.00633.
> > >
> > > **Question 1:**
> > >
> > > We agree with the reviewer that we should provide more clarity with the distinction between Tables 2 and 3. In both tables, every row is compared using the same hyperparameters and augmentations. However, Tables 2 and 3 can’t be compared against each other. In Table 2, we are using the hyperparameters for our method in which we got the best performance that is also shown in Table 5. In this case, we are performing an ablation study of using different components of the loss in unique ways. However, Table 3 you can think of it as an ablation study that we conducted with a baseline set of augmentations that also aligns with the reference [32] comparison in the table. Specifically, these choices include the symmetric augmentation scheme of SimCLR in the solo-learn library and the same LARS optimizer.  We can provide more context to make this clearer in the main paper.
> > >
> > > **Question 2:**
> > >
> > > It is true that our method is an adaptive calculation for alpha and beta. In Table 2, we provide results for the fictitious scenario in which we assume that our method is not adaptive. This means that instead of allowing alpha to vary over training, we compare it against simply setting it to a default value. In this way, it allows us to assess whether the adaptive nature of our approach is necessary. In Table 2, this is demarcated by using the “Ada” convention when we are using the default mode of our method and setting a specific value like 0 or 1 or 0.5 when we want to compare against setting these parameters to a specific number across all of training.
> > >
> > > **Question 3:**
> > >
> > > It is true that ER(Z) during training would be bounded in the batch size.  However, we note that for the purposes of training, we do not need a measure of the exact rank of the dataset as a whole. This calculation can be done during inference on the entire dataset to get a much more accurate sense of dimensionality.  Instead, what our alpha term gives us is a sense of how the dimensionality is growing across training. We observe that this growth is very specific to the dataset of interest in part c) of Figure 5. Additionally, using effective rank is very desirable for this problem because it has an upper bound (the size of the minimum dimension) that we can use to track how close we are to approaching this bound.
> > > For the purposes of the revised version of the paper, we are happy to perform an additional batch size ablation study. However, we would like to note we also experimented with changing the output embedding size in an ablation study in the appendix (part a of Figure 16) and we find that changing the matrix size does not have much of an impact on performance.

---

> ### Author Response · Authors · 2025-11-28
> **Revised Paper Update Part 1**
>
> **Weakness 1 Revised Paper Update:**
>
> We have performed experiments to improve the experimental robustness of the entire paper. This is reflected in the new Tables 3, 4, and 5. Specifically, we now have two settings: AdaDim with baseline parameters and AdaDim with parameters suitable for leveraging an additional momentum encoder model. We find that the baseline AdaDim setting can improve performance with dataset specific tuning of the gamma parameter and is comparable to methods to make use of a variety of architectural tricks. However, we also found that AdaDim leverages a momentum encoder particularly well to the point that it exhibits state of the art performance in both the solo learn and ImageNet 100 epoch benchmarks. We also analyzed the experimental robustness across different amounts of training time  in Figure 6 and the influence of a variety of hyperparameters in Figure 7.
>
> The original table 3 that had the fixed hyperparameter setting is now an ablation study in appendix Section C.9. Additionally, we have clarified the difference between the settings used in each table in the paper. In the new experimental setting, we have the baseline setting that we used in the original version of the paper and the expanded setting for hyperparameters specific to experiments with the additional momentum encoder network. All experiments are detailed fully in Section A.4 with associated pseudo code in Section A.3.
>
> **Weakness 2 Revised Paper Update:**
>
> To address this concern, we have included more contextual sentences in the introduction in order to clarify the exact context of previous work. Specifically, we have specified that previous work performed their measurements on some specific matrix related to the representation space, but not necessarily a measurement of the representation space directly.
>
> **Weakness 3 Revised Paper Update:**
>
> We have moved the original Figure 6 to appendix section C.7 and provided a new analysis in the main paper. This new Figure 6 has an analysis of the performance and dynamics of AdaDim under different amounts of training time. In this analysis figure, we show the training dynamics of different SSL methods over 2000 epochs on TinyImageNet200. Again, we see a similar trajectory between VICReg and AdaDim, but the regularization of both the NCE loss and the L_mut loss causes a decrease towards a more performance-friendly balance between H(R) and I(R;Z).
>
> **Weakness 4 Revised Paper Update:**
>
> We have added more of a discussion in the methodology section to specify that the embeddings fed into the SimCLR loss are normalized while those in the VICReg loss are unnormalized. Additionally, we have also added detailed pseudo code in Section A.3 to make the distinction more clear.
>
> **Weakness 5 Revised Paper Update:**
>
> We also find in Table 3 that tuning the gamma parameter on a dataset specific basis can also improve the performance across a  wide variety of datasets. This highlights the importance of the gamma parameter during optimization. We also provide an ablation study of varying gamma in Figure 7 and observe that specific datasets benefit from a specific degree of regularization from the L_mut term.
>
>
> **Weakness 6 Revised Paper Update:**
>
> We have provided an expanded discussion of the relation of our work with [18] at the of the related works section. We have clarified the previous interpolation comment and added a discussion regarding previous work that has hinted at differences between SimCLR and VICReg optimization.
>
> **Question 1 Revised Paper Update:**
>
> We have moved the original table 3 to supplementary section C.9 as an ablation study in the case where all methods are compared on a fixed hyperparameter basis. Instead, we have reorganized the experiments in the main paper with the baseline setting for standard AdaDim joint embedding optimization and the expanded hyperparameter setting where AdaDim is used with a momentum encoder. We have clarified these hyperparameters in section A.4 and provided comprehensive pseudocode for our method in section A.3.
>
> **Question 2 Revised Paper Update:**
>
> We added additional clarity to our writing to ensure that the reader understands that alpha and beta are adaptively computed and that Table 2 is based on a hypothetical analysis where each of these terms can be tuned individually.

---

> ### Author Response · Authors · 2025-11-28
> **Revised Paper Update Part 2**
>
> Question 3 Revised Paper Update:
>
> We provide more analysis on the parameters that influence the size of the matrix fed into the effective rank calculation in Figure 7. Specifically, we analyze the impact of the output projection size on the performance of our method. We observe that our method still out performs the baseline state of the art Resa algorithm regardless of the projection size, but AdaDim still favors a smaller projection with the best performance occurring at a size of 512. Additionally, we perform a batch size ablation study in Section C.10 and find that the resulting performance is generally better at smaller batch sizes, but is strongly dependent on the specific hyperparameters and data settings of the training setup. This all indicates that the approximate effective rank calculation we can achieve with small batches is still sufficient for the purposes of getting a sense of the dimensionality state of the model and can achieve state of the art performance even under conditions where the matrix sizes are varied.

---

### Official Review · Reviewer_oQBM · 2025-11-04

**Soundness:** 3
**Presentation:** 4
**Contribution:** 3
**Rating:** 8
**Confidence:** 4

**Summary:**

In this paper the authors study SSL (Self-supervised learning) training dynamics through the joint lens of representation dimensionality H(R) (measured via effective rank) and projection-head mutual information I(R;Z). They challenges the common view that “higher H(R) and lower
I(R;Z) are always better,” arguing instead that the best-performing models reach a balance between the two.

Using a Gaussian analysis and an information-flow bound, the authors show that early training increases H(R) via feature decorrelation which also increases I(R;Z). While later training increases H(R) mainly through sample uniformity, during which I(R;Z) plateaus or decreases.

Empirical studies on ResNet-18/50 across SimCLR, VICReg, NNCLR, BYOL, and multiple datasets corroborate these phases and the “sweet spot” relationship between H(R) and I(R;Z). Building on these insights, the paper introduces AdaDim, which

- (i) adaptively interpolates between a decorrelation loss (VICReg’s covariance term) and a sample-uniformity loss (NT-Xent) using an
α  schedule derived from the effective rank of Z, and

- (ii) gradually regularizes I(R;Z) with a Rényi-entropy-based term scaled by β=γα.

The authors show that AdaDim yields consistent improvements (up to ~3% Top-1) over matched SimCLR/VICReg setups and that AdaDim is competitive with state-of-the-art SSL methods on CIFAR-100, Tiny-ImageNet-200, CINIC-10, STL-10, and several MedMNIST datasets, while requiring only intermittent SVD computations.

**Strengths:**

- Clear dynamics story with theory. The Gaussian and information-flow analyses predict two phases (decorrelation vs. uniformity), and the measurements on real runs (eigenvalue trajectories, uniformity, matrix-MI) match those predictions, including the non-monotonic behavior of I(R;Z).

- Low-overhead method. AdaDim’s α from effective rank(Z) plus β-scaled MI regularization is easy to implement (periodic SVD on a few batches) and does not require queues, clustering, student–teacher, or extra predictors.

- Wide empirical coverage. Demonstrated consistent benefits across CIFAR-10/100, Tiny-IN-200, CINIC-10, STL-10, and MedMNIST variants; competitive ImageNet-100 results vs. strong baselines (e.g., BYOL, MoCo v3, Barlow Twins).

- Nice dissemination and comprehensive ablations. Tables/figures dissect fixed vs. cosine/linear α, with/without β, and γ sweeps; they show why adaptive schedules beat hand-crafted ones and why I(R;Z) regularization should be phased, not constant.

**Weaknesses:**

- Estimator and stability details for I(R;Z). The matrix-based Rényi-entropy estimator is used operationally, but its bias/variance, windowing, and normalization details are compactly stated. I could see a sensitivity study (batch size, feature dim, normalization choice) as beneficial.

- Theory idealization. The Gaussian joint model and the assumption that I(Y;Z) approaches a constant simplify the narrative; stronger connections to non-Gaussian encoders/projectors or a small-scale non-linear toy model (beyond PCA) would further shore up generality. (Some neural projector simulations are in the appendix; surfacing them in main text might also be helpful.)

- Scope of SOTA comparisons. While solo-learn baselines are strong, results focus on online linear eval and ~400-epoch training. Maybe a few shorter training regimes and transfer evals (linear probe on external datasets; few-shot) can further stress-test the “balance” claim.

- Hyperparameter coupling. β=γα ties MI regularization strictly to α. Some datasets may benefit from asynchronous schedules (e.g., a later spike in MI reg while α stabilizes). An ablation where β lags α could uncover improvements.

**Questions:**

- Estimator sensitivity. How sensitive are conclusions to the matrix-MI estimator choice (Rényi order, normalization, batch size)? Have the authors quantitative results about variance across seeds/batches and any bias relative to kNN-MI or MINE on low-dim toy data?

- Asynchronous schedules. Have the authors tried decoupling β from α (e.g., delayed ramp or piecewise schedule)? Can the authors explain more on that and/or add a plot of (H(R),I(R;Z)) trajectories under a few schedule families could validate the “balanced-path” intuition?

- Short-train regimes. The authors note that a fixed I(R;Z) regularization seems tied to 400-epoch runs. What happens at 100–200 epochs (where many practitioners operate)? Does AdaDim still outperform fixed-λ MI reg?

- Transfer and robustness. Beyond online linear eval, can the authors elaborate/explain how would the learned balance correlate with transferability (linear probes result on out-of-domain datasets or few-shot finetuning would be nice t have)?

- Compute overhead. Would be nice to see a wall-clock overhead breakdown for the SVD step (per epoch %) across datasets/batch sizes, and any approximate alternatives (e.g., randomized SVD or Nyström).

- Failure modes. Are there datasets where AdaDim’s α quickly saturates to ~1 (or sticks near 0) and harms training? Could the authors elaborate whether a capped or layer-wise effective-rank signal would help there in such case.

**Details Of Ethics Concerns:**

N/A.

---

> ### Author Response · Authors · 2025-11-15
> **Initial Response to Weaknesses and Questions Part 1**
>
> Thank you reviewer oQBM for taking the time to evaluate our work! We look forward to in-depth discussions over the next few weeks.
>
> We would also like to point out that we are currently working on the revision that will be uploaded soon.
>
> **Weakness 1:**
>
> We agree that a study of different common hyperparameters would be beneficial for experimental robustness. In Appendix C, we have studies for the projection output size, temperature variation, and rate of number of updates to the alpha parameter. Specifically, Figures 15 and 16 discuss hyperparameter sensitivity. We are running experiments to address ablation studies regarding batch size and normalization. We will reply to this comment with the new table and upload the revised paper with the ablationsl.
>
> **Weakness 2:**
>
> We agree that including more of a comprehensive look at the PCA projection might help with generality. This neural network simulation is currently in the appendix and space permitting we may bring it up into the main paper as well. We will consider this point and potentially bring up the neural network simulation into the main paper as well.
>
> **Weakness 3:**
>
> We agree that additional experimental robustness experiments could help with the generality of our method. Specifically, we will explore experiments on longer and shorter training regimes and analyze the impact of these choices in the revised version of the paper for the rebuttal.
>
> **Weakness 4:**
>
> We agree that potentially more performance improvements can be obtained with dataset specific tuning of our alpha and beta relationship. For the sake of generality, we focus on maintaining the fixed beta = gamma * alpha relationship, but it is absolutely true that certain datasets exhibit different dynamics that may benefit from asynchronous schedules between each of these terms. We can potentially explore these relationships to some degree in a revised version of the paper for the rebuttal.
>
> **Question 1:**
>
> In Appendix C, we have studies for the projection output size, temperature variation, and rate of number of updates to the alpha parameter. Specifically, look at Figures 15 and 16 that provide a discussion of hyperparameter sensitivity. We will definitely look at the impact of other parameters that you suggested such as batch size and renyi order on downstream performance and provide an update in a revised version of the paper.
>
> **Question 2:**
>
> In Table 2, we provide ablation studies where the alpha and beta parameters are partially decoupled from each other. In this case, we have one row where the alpha parameter is set adaptively, but the beta parameter is a constant throughout training. This seems to perform worse than our adaptive approach. The main reason we haven’t tried decoupling schedules is that we performed ablation studies on alpha where we set a cosine or linear schedule that consistently performs worse than our rank based schedule. We also prefer to have a method with fewer heuristic hyperparameters such as setting a cosine or linear schedule. However, these ablation studies could be very interesting and we can definitely provide more training trajectory plots under different conditions to get a sense of these trends.
>
> **Question 3:**
>
> We need to test these short term training regimes. We will have this as part of our ablation studies for the revised version of the paper. I suspect that we will still observe performance improvements compared to the fixed version simply because our method can adapt to the dynamics of these settings as well, but it will be an interesting study that we can include in the revised version of this paper.
>
> **Question 4:**
>
> A major intuition regarding our method is the relationship between information content and noise. A toy figure regarding this difference can be found in section B.1 of the appendix. The idea behind regularizing both H(R) and I(R;Z) is that high H(R) is good, but it implies that we are learning features that are potentially not useful such as noisy components in images that don’t correlate to semantic concepts. By regularizing both H(R) and I(R;Z), we are trying to find that balance point where we learn useful information, but at the same time screen out potential noise that won’t contribute to any downstream task. For this reason, I believe that our method will perform particularly well on downstream out of distribution tasks simply because it has a mechanism to balance between information and noise content. This is displayed by our better results across both traditional classification datasets as well as more imbalanced datasets such as the medmnist dataset. This may help its performance to a greater degree in out of distribution settings.

---

> > ### Author Response · Authors · 2025-11-15
> > **Initial Response to Weaknesses and Questions Part 2**
> >
> > **Question 5:**
> >
> > This is a good ablation study suggestion and we can definitely do a compute overhead comparison in some manner.
> >
> >
> > **Question 6:**
> >
> > I have not seen a case where the alpha parameter of AdaDim approaches a full 1. In part c) of Figure 5, we show how the alpha term evolves during training for a different datasets. As we showed in the training dynamics section of the paper, at some point during training the model arrives at a point where it struggles to project into higher dimensional spaces. This point is different for every dataset and is a major motivator behind our work that can adapt these dynamics to the characteristics specific to the data regime of interest. However, some data specific scenarios may benefit from scaling the alpha parameter with effective rank at a much slower or faster rate than we do in our work. For example, noisy datasets may require a greater degree of I(R;Z) regularization to force the model to only focus on components that are relevant for the downstream task. In such a case, I could see benefit from rapidly increasing alpha. This is an interesting point for future work. We also have not explored how the trajectory of rank along intermediate layers affects this regularization. In general, much of the SSL literature revolves around the encoder and the projector and there is less understanding of the behavior of intermediate states in a model. It would be interesting to see if our calculations could also be influenced by the growth in rank of intermediate representations at different points in the encoder.

---

> ### Author Response · Authors · 2025-11-28
> **Revised Paper Discussion**
>
> **Weakness 1 Revised Paper Update:**
>
> We have included a comprehensive parameter ablation study in Figure 7 of the paper and a complete batch size discussion in Section C.10.
>
> **Weakness 2 Revised Paper Update:**
>
> We have provided additional justification for the usage of PCA as an analytical projection in the main paper and tried to highlight the neural network simulation to a greater degree. However, we did not end up having the space necessary to move this simulation to the main paper.
>
> **Weakness 3 Revised Paper Update:**
>
> We have performed additional experiments to improve the experimental robustness of the entire paper. This is reflected in the new Tables 3, 4, and 5. Specifically, we now have two settings: AdaDim with baseline parameters and AdaDim with parameters suitable for leveraging an additional momentum encoder model. We find that the baseline AdaDim setting can improve performance with dataset specific tuning of the gamma parameter and is comparable to methods to make use of a variety of architectural tricks. However, we also found that AdaDim leverages a momentum encoder particularly well to the point that it exhibits state of the art performance in both the solo learn and ImageNet 100 epoch benchmarks. We also analyzed the experimental robustness across different amounts of training time  in Figure 6.
>
>
> **Weakness 4 Revised Paper Update:**
>
> We analyzed performance improvements as the gamma performance is tuned in Table 3. We find that dataset specific tuning for individual datasets results in better performance compared to the original setting where all datasets receive the same gamma parameter of 1e-4.
>
> **Question 1 Revised Paper Update:**
>
> We have included a comprehensive parameter ablation study in Figure 7 of the paper and a complete batch size discussion in Section C.10.
>
> **Question 2 Revised Paper Update:**
>
> We believe these are good directions for future studies. Thank you for the interesting discussion on this point!
>
> **Question 3 Revised Paper Update:**
>
> We provide additional analysis in short and long term training regimes in Figure 6.
>
> **Question 4 Revised Paper Update:**
>
> As discussed previously, we believe our method may be well suited for an OOD setting as it shows performance improvements on datasets within a wide variety of settings and not just on the ImageNet benchmark. These performance improvements are further demonstrated through the introduction of the momentum encoder network.

---

> > ### Author Response · Authors · 2025-12-01
> > **Revised Paper Discussion Part 2**
> >
> > We have added a computational cost comparison analysis in Section A.8.

---

### Author Response · Authors · 2025-11-28
**Revised Rebuttal Version of the Paper Available Part 1**

**Overall Response to Paper Additions**

To organize the discussion of changes made to the main paper, we will discuss them with different sections that reflect the type of feedback provided by the reviewers.

Note that we also tried to highlight in blue text any changes that we felt were specifically included to address reviewer feedback.

Note that further changes may be made until the December 3rd deadline, but we wanted to make sure an initial version reached the reviewers with sufficient time for them to go through the revised work thoroughly.

**Experimental Feedback:**

The most significant improvement in this version of the paper is greater experimental robustness across a variety of application settings.

Specifically, we introduce the baseline hyperparameter setting and the expanded hyperparameter setting. The baseline setting is the basic joint embedding architecture that we used in the original paper. The expanded hyperparameter setting is the set of hyperparameters used when we integrate AdaDim with a momentum encoder. In this setting, we use the training hyperparameters of [1] which is the most recent state of the art approach that utilizes a momentum encoder during training. We show the performance of AdaDim in both the original setting that does not use a momentum encoder network and the expanded setting that utilizes momentum encoding. Note that many other SSL approaches use momentum encoders in tandem with their standard backbone network. This is not something that affects the design of the loss function, but just impacts the way that representations are updated through the SSL methodology.

For additional clarity on these changes, we provide detailed pseudocode and training details of AdaDim in Section A.3 and A.4. We observe that training with the momentum encoder leads to significant margins of improvement over other state of the art approaches in Tables 3 ,4, and 5. More significantly, we now show significant performance gains over previous SSL  benchmarks which includes the solo-learn benchmark in Table 4 and the 100 epoch ImageNet training benchmark in Table 5. We also highlight the architectural strategies used in all our comparison methods. We note that even without the momentum encoder or any other additional SSL strategies, AdaDim out performs many other methods. However, when using the additional momentum encoder AdaDim has a significant margin of improvement over everything else, including [1] where we mimic the same training hyperparameters as much as possible.

We also have new robustness experiments based on reviewer feedback. This includes a comprehensive hyperparameter analysis in Figure 7, a training epoch analysis in Figure 6, a compute cost analysis in Section A.8, a batch size analysis in Section  C.10, and a vision transformer analysis in Section C.8.

To prioritize the feedback of the reviewers in the main paper, we moved certain ablation studies that were originally in the main paper to the appendix. This includes the original Figure 6 that was an analysis of the gamma parameter and is now its own separate appendix section C.7. Additionally, the original Table 3 that was an analysis of certain choices of the loss function in a fixed hyperparameter setting is now its own separate Section C.9.

[1] Weng, X., An, J., Ma, X., Qi, B., Luo, J., Yang, X., ... & Huang, L. (2025). Clustering Properties of Self-Supervised Learning. arXiv preprint arXiv:2501.18452.

**Discussion Feedback:**

We have also integrated additional discussion points into the main paper and appendix. These are represented by blue highlighted sentences throughout the paper. Specifically, we include further discussions to contextualize our approach against previous approaches in both the introduction and related works sections. We have also updated our discussion in Section C.4 to further build on points made by the reviewers.

We have also taken additional lengths to justify the usage of PCA and a gaussian simulation in our analysis of Section 3. These additional details are highlighted in the main text and further discussed in Sections B.2, B.3, and B.4. Note that we try to highlight that our neural network simulation also leads to the same conclusions as those of this gaussian analysis. Furthermore, we provide more details regarding our analysis in Figure 3 with a complete discussion of this figure in Section A.9.

---

> ### Author Response · Authors · 2025-11-28
> **Revised Rebuttal Version of the Paper Available Part 2**
>
> **Clarity Feedback:**
>
> We have taken a variety of approaches to improve the overall clarity of the paper. This includes much more experimental details section in A.4. Additionally, we now provide a complete pseudocode of our AdaDim algorithm in section A.3. We have also attempted to improve the discussion of our methodology. We highlight specifically the normalization of embeddings for the NCE loss, complete details of the NCE and VICReg losses in Section A.5, Section A.6 where the mutual information calculation can be found, and additional details to clarify the usage of the momentum encoder network.
>
> Furthermore, the paper now has more references to specific sections in the appendix where useful details are located in an effort to make the paper as self contained as possible.

---

> ### Author Response · Authors · 2025-12-01
> **Additional Results to Revised Version of the Paper**
>
> In Table 3, we have added more experimental results to compare our AdaDim method against the most recent start of the art Resa algorithm that uses  a momentum encoder alongside a skinkhorn knopp computation step.
>
> In this table, we train the Resa algorithm across a wide variety of data settings us the expanded hyperparameter setup that was proposed in their original paper.
>
> We find that our AdaDim + momentum encoder setup consistently out performs the Resa algorithm across the vast majority of datasets.

---

### Author Response · Authors · 2025-12-01
**Summary of Revision Discussion for Area Chair**

We sincerely thank the Area Chair for taking the time to perform an additional assessment of our work. To help summarize all the discussions that took place so far, we provide a brief overview of our contributions, the reviewers’ main concerns, and the updates we made to address them.

# **Contributions**

All reviewers noted that our proposed characterization of SSL training dynamics based on info theoretic terms is an interesting and novel contribution. Additionally all reviewers noted that our method (AdaDim) was comparable to or out performed competing strategies without additional architectural or computational overhead. Furthermore, reviewers enjoyed how the construction of our loss connected back to the novel training dynamics we introduced.

# **Reviewer Specific Analysis**

While reviewers consistently acknowledged the novelty of our training dynamics and the strong points regarding AdaDim, their main concerns centred on  verifying the experimental robustness of our method, particularly on state of the art benchmarks and within the context of hyperparameter sensitivity. Certain reviewers also wanted clarification regarding the relationship with previous literature and the notations used when describing the method.

### **Reviewer oQBM (Score: 8, Confidence: 4)**

**Initial Critique:**

This reviewer supported many aspects of the paper, but they noted that further experiments would help with validating the robustness of the method.

**Our Solution:**

We included Figure 6 which is an analysis of our method across different amounts of training time as well as the training trajectories for different methods trained in this setting. We also included other experiments this reviewer requested such as hyperparameter sensitivity experiments, compute cost experiments, and batch size experiments.

### **Reviewer MV6A(Score: 4, Confidence: 4)**

**Initial Critique:**

This reviewer seemed very willing to increase their score as stated directly in their review, but they wanted to see more experiments, particularly on more standard benchmarks (ImageNet, solo-learn) and hyperparameter robustness studies.

**Our Solution:**

We demonstrated that with proper tuning AdaDim achieves state of the art performance on the solo-learn benchmark (Table 4), the ImageNet benchmark (Table 5), and when compared against the most recent state of the art Resa algorithm (Table 3,4,5). We also demonstrate robustness against many different hyperparameters in Figure 7.

### **Reviewer ACRX (Score: 2, Confidence: 4)**

**Initial Critique:**

This reviewer questioned the experimental robustness of our method as well as its exact relationship with previous work. The reviewer also questioned the exact impact of certain choices of our method on the performance increases we observed.

**Our Solution:**

We provided an in depth analysis of both our work and previous work in order to contextualize our contribution in both the related works and introduction. Additionally, we provide, in Figures 6 and 7, a comprehensive analysis of hyperparameter sensitivity and training dynamics of our method. Finally, we provide state of the art improvements of our method in Tables 3,4, and 5.

### **Reviewer ZhAj (Score: 2, Confidence: 3)**

**Initial Critique:**

This reviewer’s main criticism seemed to be the clarity of the paper with problems such as notations, choice of analytical setting, and understanding where certain conventions came from.

**Our Solution:**

We revamped several locations in the paper in order to bring out the clarity of various details. This includes a full detailed pseudocode of our method, more explicit language for the usage of certain conventions like the PCA analysis, and where to locate specific details such as the fully worked out equation for the VICReg loss.

---

### Meta-Review · Area_Chair_iVJi · 2025-12-07

**Summary:**

Reviewers appreciated the paper’s information-theoretic view of SSL dynamics and the lightweight AdaDim scheme, but raised several major concerns:

1. The reviewer questioned how novel AdaDim really is beyond an adaptive combination of SimCLR/VICReg, and whether the added loss terms and hyperparameter γ are justified by only modest gains in fully tuned settings.

2. The reviewers worried about theoretical idealizations (Gaussian/PCA analysis, assumptions on I(Y;Z)) and the limited initial scope of experiments (small/medium datasets, no batch-size/temperature/compute studies, limited transfer).

3. Two reviewers found the original exposition unclear, with missing notation, imprecise positioning w.r.t. prior work, and cramped figures/tables.

**Reviewer Concerns:**

**Addressed concerns**: the revision adds state-of-the-art results on solo-learn and ImageNet, extensive hyperparameter and batch-size ablations, compute-cost analysis, ViT experiments, clearer notation and pseudocode, and a more careful discussion of prior work and PCA/Gaussian assumptions.

**Remaining concerns**: lingering doubts about conceptual novelty vs existing SSL theory, reliance on idealized estimators/assumptions, and whether AdaDim’s extra complexity is fully warranted by its performance margins.

The AC also noted that the paper is highly related to “LDReg: Local Dimensionality Regularized Self-Supervised Learning,” and that the reference formatting does not follow the standard ICLR style.

Overall, the AC recommends rejection of the paper. Nevertheless, the AC encourages the authors to refine the clarity, strengthen experimental validation, and resubmit, as the underlying ideas show promising potential.

**Reviewer Scores:**

oQBM: 8
ACRX: 2
ZhAj: 2
MV6A: 4

---

### Decision · Program_Chairs · 2026-01-26

Reject